# Causal Identification from Counterfactual Data: Completeness and Bounding Results

**Arvind Raghavan** [* 1]

## Abstract

Previous work establishing completeness results for *counterfactual identification* has been circumscribed to the setting where the input data belongs to observational or interventional distributions (Layers 1 and 2 of Pearl's Causal Hierarchy), since it was generally presumed impossible to obtain data from counterfactual distributions, which belong to Layer 3. However, recent work (Raghavan & Bareinboim, 2025) has formally characterized a family of counterfactual distributions which can be directly estimated via experimental methods - a notion they call *counterfactual realizability*. This leaves open the question of what *additional* counterfactual quantities now become identifiable, given this new access to (some) Layer 3 data. To answer this question, we develop the CTFIDU$^+$ algorithm for identifying counterfactual queries from an arbitrary set of Layer 3 distributions, and prove that it is complete for this task. Building on this, we establish the theoretical limit of which counterfactuals can be identified from physically realizable distributions, thus implying the *fundamental limit to exact causal inference in the non-parametric setting*. Finally, given the impossibility of identifying certain critical types of counterfactuals, we derive novel analytic bounds for such quantities using realizable counterfactual data, and corroborate using simulations that counterfactual data helps tighten the bounds for non-identifiable quantities in practice.

---

[*]Elias Bareinboim (eb@cs.columbia.edu) was erroneously omitted from the list of authors. [1]Causal Artificial Intelligence Lab, Department of Computer Science, Columbia University. Correspondence to: Arvind Raghavan <ar@cs.columbia.edu>.

*Proceedings of the 43$^{rd}$ International Conference on Machine Learning*, Seoul, South Korea. PMLR 306, 2026. Copyright 2026 by the author(s).

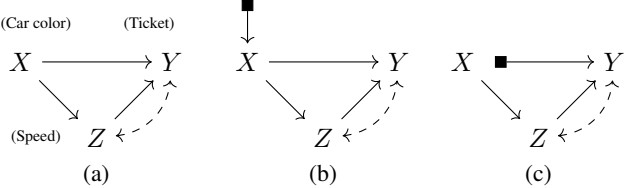

*Figure 1.* (a) Causal diagram for Ex. 1 (Traffic Camera); (b) Standard randomization overriding $X$ and affecting both $Z, Y$; (c) Counterfactual randomization of $X$ affecting $Y$, but not $Z$.

## 1. Introduction

The Pearl Causal Hierarchy (PCH) provides a foundational framework for reasoning about causality (Pearl & Mackenzie, 2018; Bareinboim et al., 2022). The hierarchy formalizes three progressively richer modes of reasoning—*seeing*, *doing*, and *imagining*—which correspond to *observational*, *interventional*, and *counterfactual* regimes within an environment of interest. Consider the following example:

**Example 1 (Traffic Camera).** Consider a fairness auditor reviewing an AI system for issuing speeding tickets based on traffic footage. $X$ represents the color of the car, $Z$ the driving speed, $Y$ the decision to issue a ticket. Fig. 1a shows the auditor's causal graphical assumptions: due to a high correlation in the training data between the speeding tendencies and car-color preference of different socioeconomic groups, $X$ might directly affect $Y$ in the algorithm. $X$ might affect $Z$ if pedestrians and other drivers react to, say, a red car and affect its speeding. Speeding and outcome might be affected by an unobserved confounder: unlabeled road obstacles (which present as video artifacts). □

The first layer of the PCH ($\mathcal{L}_1$) captures *observational* distributions such as $P(Y = 1 \mid X = x)$, how likely are drivers of $x$-colored cars to receive a ticket. The second layer ($\mathcal{L}_2$) concerns *interventional* distributions, such as $P(Y_x = 1)$, how likely is a speeding ticket when car color is fixed as $x$, say, by an experiment recruiting drivers and randomly assigning them test cars, as shown in Fig. 1b. The third layer ($\mathcal{L}_3$) addresses *counterfactual* distributions over conflicting realities, for example $P(Y_x = 1 \mid X = x')$, the probability a driver receives a ticket if assigned an $x$-colored car, given that the original color was $x'$. Higher layers subsume lower layers. It is well-established that higher-layer questions can-

*Table 1.* Comparison of different algorithms for counterfactual identification and the scope of input data. Ours is complete when assuming an arbitrary set of physically realizable input data.

| Method | ID Query | Input Data |
|---|---|---|
| IDC* | $\mathcal{L}_3$ query | Full $\mathcal{L}_2$ data |
| PSIDC | Path-specific $\mathcal{L}_3$ | Full $\mathcal{L}_2$ data |
| CTFID | $\mathcal{L}_3$ query | Subset of $\mathcal{L}_2$ data |
| **CTFIDU$^+$** (ours) | $\mathcal{L}_3$ query | Subset of realizable $\mathcal{L}_3$ data |

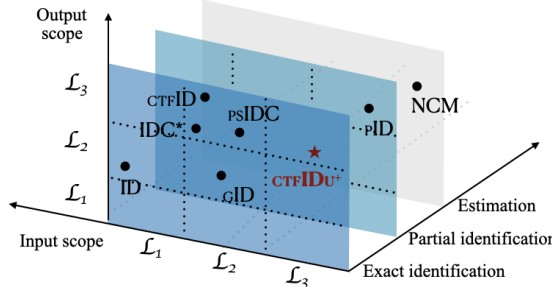

*Figure 2.* Landscape of causal identification and estimation methods. CTFIDU$^+$ is complete for $\mathcal{L}_3$ identification from a collection of realizable counterfactual data.

not be answered using data from lower layers alone, and require causal assumptions to perform inference (Ibeling & Icard, 2020; Bareinboim et al., 2022).

Counterfactuals are widely acknowledged to be important in topics including personalized decision-making (Bareinboim et al., 2015; Mueller & Pearl, 2023), path-specific effect estimation (Pearl, 2001; Rubin, 2004; Avin et al., 2005), fairness analysis (Zhang & Bareinboim, 2018; Plecko & Bareinboim, 2024), explainable AI (Lee et al., 2025) etc. This has spurred much work in the field of counterfactual *identification* (defined in Sec. 2): Shpitser & Pearl (2008) proved their IDC* algorithm is complete for identifying an $\mathcal{L}_3$ quantity when assuming knowledge of all $\mathcal{L}_2$ (inc. $\mathcal{L}_1$) data. Using this result, Malinsky et al. (2019) developed the PSIDC algorithm for path-specific effect identification. Correa et al. (2021) then proved their CTFID algorithm is complete for $\mathcal{L}_3$ identification, assuming access to a *subset* of $\mathcal{L}_2$ data, and Correa et al. (2022) extended this to counterfactual *transportability* across heterogeneous environments. If a counterfactual is non-identifiable, Zhang et al. (2022) provide a Bayesian sampling method, which we call PID, to *partially* identify the bounded range of this quantity. These methods are depicted in Table 1 and Fig. 2, along with the dimensions of consideration: which quantities the method is capable of identifying (output scope) and the scope of input data it assumes. We also distinguish between identification methods, which map counterfactuals to a unique function of input data, and statistical methods for practically estimating these functions using finite samples of input data.

To appreciate the relevance of counterfactuals, consider the *natural direct effect* (NDE) of a treatment $X$ on outcome $Y$ (Pearl, 2001). NDE is defined as $P(Y_{xZ_{x'}} = 1) - P(Y_{x'} = 1)$, where the first term is a *nested* counterfactual. In Ex. 1, $P(Y_{xZ_{x'}} = 1)$ denotes the outcome probability if a driver's car were randomly assigned color $x$ *and* speeding $Z$ was fixed to what it *would have been* had her car been assigned color $x'$. Decomposing the total effect of $X$ on $Y$ this way allows an auditor to reason about algorithmic fairness in this scenario. Unfortunately, due to unobserved confounding in the graph in Fig. 1a, NDE is non-identifiable using only $\mathcal{L}_2$ data. Hence, the algorithms cited earlier fail to identify it, since they assume the input data is limited to $\mathcal{L}_2$.

It is commonly believed that $\mathcal{L}_3$ distributions are inaccessible except indirectly via identification (e.g., see Dawid, 2000; Shpitser & Pearl, 2007). However, Raghavan & Bareinboim (2025) recently provided a formal characterization of a family of counterfactuals which *can* be directly sampled from in an experimental setting, a property they term *counterfactual realizability*. This is made possible by the discovery of a physical procedure called *counterfactual randomization* (Bareinboim et al., 2015), which permits $\mathcal{L}_3$ data collection. E.g., in Ex. 1, the auditor can randomize the RGB values in the video footage to fix the car color $X$ *as perceived* by $Y$, without affecting the natural value of $X, Z$, as shown in Fig. 1c. NDE can be identified with this data as follows:

$$P(Y_{xZ} = 1 \mid X = x') \qquad \text{by ctf. randomization} \quad (1)$$
$$= P(Y_{xZ_{x'}} \mid X = x') \qquad X = x' \implies Z = Z_{x'} \quad (2)$$
$$= P(Y_{xZ_{x'}}) \qquad \qquad \text{d-separation} \quad (3)$$

The NDE now becomes identifiable with the possibility of counterfactual data collection. This realization in fact opens up more fundamental questions: *which other $\mathcal{L}_3$ quantities also become identifiable given access to (some) $\mathcal{L}_3$ data? What is the relationship between counterfactual identifiability and realizability – does one imply the other?* We resolve these questions in this paper.

Specifically, our contributions are as follows:
- Sec. 3: we develop the CTFIDU$^+$ algorithm (Alg. 2) which identifies a counterfactual quantity using data from an arbitrary set of $\mathcal{L}_3$ input (inc. realizable counterfactual data), or returns FAIL if the query is non-identifiable. We prove the algorithm is complete (Thm. 3.5). CTFIDU$^+$ thus subsumes the previous algorithms in Table 1.
- Sec. 4: we prove foundational results connecting counterfactual realizability and identifiability. We show that the theoretical limits of realizability are also the limits of exact identification (Thm. 4.1). This further implies a duality result: a counterfactual quantity is identifiable iff its distribution is realizable, in principle, via counterfactual randomization actions (Cor. 4.2).

- Sec. 5: we show that, even for non-identifiable quantities, the partial identification bounds can be tightened by accessing (some) counterfactual data. We derive novel analytic bounds for an important type of $\mathcal{L}_3$ query using counterfactual data which are provably tighter than previous results (Prop. 5.4). We then show via simulations that this extra data meaningfully narrows the $(1 - \beta)$ credible interval for an identification query in practice (Ex. 2, 3).

For proofs of all the results, refer to the full technical report (Raghavan & Bareinboim, 2026).

## 2. Background and Notation

We denote variables by capital letters, $X$, and values by small letters, $x$. Bold letters, $\mathbf{X}$, are sets of variables and $\mathbf{x}$ sets of values. $P(\mathbf{x})$ is shorthand for $P(\mathbf{X} = \mathbf{x})$. $\mathbb{1}[.]$ is the indicator function. Two values $\mathbf{x}$ and $\mathbf{z}$ are consistent if they share the common values for $\mathbf{X} \cap \mathbf{Z}$. We denote by $\mathbf{x} \setminus \mathbf{Z}$ the subset of $\mathbf{x}$ corresponding to variables in $\mathbf{X} \setminus \mathbf{Z}$, and by $\mathbf{x} \cap \mathbf{Z}$ the subset of $\mathbf{x}$ corresponding to variables in $\mathbf{X} \cap \mathbf{Z}$. We assume finite-domain discrete variables.

**Structural Causal Model.** We use *Structural Causal Models* (SCMs) to describe the generative process for a system (Bareinboim, 2025; Pearl, 2000). An SCM $\mathcal{M}$ is a tuple $\langle \mathbf{V}, \mathbf{U}, \mathcal{F}, P(\mathbf{u}) \rangle$. $\mathbf{V}$ is the set of observable variables in the system. $\mathbf{U}$ is the set of unobservable variables exogenous to the system, distributed according to $P^{\mathcal{M}}(\mathbf{U})$. $\mathcal{F} = \{f_i\}$ is a set of functions s.t. each $f_i$ causally generates the value of $V_i \in \mathbf{V}$ as $V_i \leftarrow f_i(\mathbf{U}_i, \mathbf{Pa}_i)$, where $\mathbf{U}_i \subseteq \mathbf{U}$ and $\mathbf{Pa}_i \subseteq \mathbf{V} \setminus V_i$. $\mathcal{M}$ is typically unknown.

**Causal diagram.** Each $\mathcal{M}$ induces a *causal diagram* $\mathcal{G}$, which is a graph containing a vertex for each $V_i \in \mathbf{V}$, a directed edge from each node in $\mathbf{Pa}_i$ to $V_i$, and a bidirected edge between $V_i, V_j$ if $\mathbf{U}_i, \mathbf{U}_j$ are not independent. $\mathcal{G}_{\overline{\mathbf{X}}\underline{\mathbf{W}}}$ denotes the result of removing edges coming into variables in $\mathbf{X}$, and edges coming out of $\mathbf{W}$. $\mathcal{G}[\mathbf{W}]$ denotes a subgraph of $\mathcal{G}$, which includes only $\mathbf{W}$ and the edges among its elements. We use standard terminology like parents, descendants of a node (see App. A). Our treatment is limited to *recursive* SCMs, which implies acyclic diagrams.

Given graph $\mathcal{G}$, its vertices can be partitioned into *confounded, or c-components* such that two variables belong to the same c-component if they are connected in $\mathcal{G}$ by a path made entirely of bidirected edges.

**Potential response.** The $do(\mathbf{x})$ operator indexes a submodel $\mathcal{M}_{\mathbf{x}}$ where the functions generating $\mathbf{X}$ are replaced with constant values $\mathbf{x}$. I.e., this is an intervention in the model $\mathcal{M}$ which overrides natural mechanisms and assigns fixed values $\mathbf{x}$ to variables $\mathbf{X}$. A variable $Y \notin \mathbf{X}$ evaluated in this regime is called a *potential response*, denoted $Y_{\mathbf{x}}$.

**Layers of the PCH.** $(\mathbf{W}_\star = \mathbf{w})$ denotes an arbitrary counterfactual event, e.g. $(Y_x = y, Y_{x'} = y', X = x'')$ denotes the joint realization of these "cross-regime" potential responses for a single unit in the study population. $\mathbf{V}(\mathbf{W}_\star)$ denotes the observable variables appearing in $\mathbf{W}_\star$, e.g. $\{Y, X\}$ in the preceding. The probability of this event $P(\mathbf{W}_\star = \mathbf{w})$ is given by the *Layer 3 ($\mathcal{L}_3$) valuation*:

$$\sum_{\mathbf{u}} \left( \prod_{W_{\mathbf{t}} \in \mathbf{W}_\star} \mathbb{1}[W_{\mathbf{t}}(\mathbf{u}) = w] \right) P(\mathbf{u}), \qquad (4)$$

with $w$ taken from $\mathbf{w}$. If the subscripts of all the terms in $\mathbf{W}_\star$ are the same $\mathbf{x}$, this corresponds to the Layer 2 ($\mathcal{L}_2$) distribution $P(\mathbf{W}_{\mathbf{x}}) = P(\mathbf{W}; do(\mathbf{x}))$. If the subscripts are all $\varnothing$, this is the Layer 1 ($\mathcal{L}_1$) distribution $P(\mathbf{W})$. We assume throughout that all distributions are positive.

$\mathbf{W}_\star$ could include potential responses under recursively defined regimes (Correa & Bareinboim, 2025, Sec. 2.1.1). For instance, in Fig. 1, the *nested counterfactual* $Y_{xZ_{x'}}$ refers to the variable $Y$ measured in a regime where $X$ is fixed to be $x$, and $Z$ is fixed to the value it would have taken had $X$ been fixed as $x'$. Such nesting can be arbitrarily deep.

**Counterfactual (ctf-) factor.** Let $\mathbf{C}_\star$ be a counterfactual set of the form $\{V_{1_{[\mathbf{pa}_1]}}, ..., V_{k_{[\mathbf{pa}_k]}}\}$, and $\mathbf{c} = \{v_1, ..., v_k\}$, with $V_i \in \mathbf{V}$. Then, $Q[\mathbf{C}_\star](\mathbf{c})$ is called the *counterfactual, or ctf-factor* of $\mathbf{C}_\star$ and is defined as

$$Q[\mathbf{C}_\star](\mathbf{c}) = P(\mathbf{C}_\star = \mathbf{c}), \qquad (5)$$

This is a generalization of the $\mathcal{L}_2$ notion of a *confounded, or c-factor*, defined for $\mathbf{C} \subseteq \mathbf{V}$ and $\mathbf{c} \subseteq \mathbf{v}$ as

$$Q[\mathbf{C}](\mathbf{v}) = P(\mathbf{c}; do(\mathbf{v} \setminus \mathbf{c})) \qquad (6)$$

**Counterfactual identification.** A query $P(\mathbf{Y}_\star = \mathbf{y})$ is said to be *identifiable* from a set of input data distributions $\mathbb{A}$ given causal diagram $\mathcal{G}$, if $P(\mathbf{Y}_\star = \mathbf{y})$ is uniquely computable from $\mathbb{A}$ in any causal model which induces $\mathcal{G}$.

**Counterfactual randomization.** (Raghavan & Bareinboim, 2025, Def. 2.3) Given a graph $\mathcal{G}$, this intervention assigns the value of a treatment variable $X$, *as perceived* by some of its child variables $\mathbf{C} \subseteq \mathbf{Ch}(X)$, to be a randomly assigned value, notated *ctf-rand($X \rightarrow \mathbf{C}$)*. Unlike the standard randomized intervention *rand($X$)*, this neither (1) overrides the unit's naturally realized value of $X$, nor (2) affects the variables in $\mathbf{Ch}(X) \setminus \mathbf{C}$. For instance, in Fig. 1c, the action *ctf-rand($X \rightarrow Y$)* affects only $Y$ without affecting $Z$, and does not override the natural $X$.[1]

---

[1] Refer to (Raghavan & Bareinboim, 2025, App. E) for the conditions that permit such a procedure to be performed.

# 3. Identification from Counterfactual Data

As discussed in previous sections, the possibility of performing *ctf-rand()* expands the scope of input data available for supporting counterfactual identification. We index each input data distribution intuitively by the actions $\mathcal{A}$ an experimenter takes in that data-collection regime. For instance, in Fig. 1c, the input distribution is indexed by the action set $\mathcal{A} = \{$ *ctf-rand(X $\rightarrow$ Y)* $\}$. Here, the experimenter is able to sample directly from the counterfactual distribution $P(Y_{xZ} = y, Z = z, X = x')$ which by the consistency rule is equivalent to $P(y_{xz}, z, x')$. The set of available data distributions indexed by $\mathbb{A} = \{\mathcal{A}_1, ..., \mathcal{A}_k\}$ thus forms an input to our identification algorithm detailed next.[2]

Our first contribution is the IDENTIFY[+] algorithm (Alg. 1), which takes as input a ctf-factor $Q[\mathbf{T}_\star](\mathbf{t})$ that can be obtained from the input data, and computes the value of some other target ctf-factor $Q[\mathbf{C}_\star](\mathbf{c})$ which is a subset ($\mathbf{c}_\star \subseteq \mathbf{t}_\star$) iff it is identifiable from this input data. For instance, in Fig. 6c, suppose we can access the $P(y_x, x')$ distribution by the action of *ctf-rand(X $\rightarrow$ Y)*, and we want to compute $P(y_x)$ using this input data. Calling IDENTIFY[+]$(\mathcal{G}, P(y_x), P(y_x, x'))$ defines $\mathbf{H}_\star := Y_x$ in Line 3. And Line 4 returns $P(y_x) = \sum_{x'} P(y_x, x')$ as needed.[3] Notably, IDENTIFY[+] generalizes the celebrated IDENTIFY algorithm (Tian & Pearl, 2003, Sec. 4.4), which works at the level of interventional ($\mathcal{L}_2$) c-factors.

In order to build up to the completeness of IDENTIFY[+] for this task, we formulate two novel data structures involving potential responses: *counterfactual forests* and *hedges*.

**Definition 3.1** (Counterfactual (Ctf-) Forest). Let $Q[\mathbf{T}_\star](\mathbf{t})$ be a ctf-factor satisfying the following:

  i. $V_{j[\cdot]}$ appears at most once in $\mathbf{T}_\star = \{V_{1_{[\mathbf{pa}_1]}}, ..., V_{k_{[\mathbf{pa}_k]}}\}$ for any $j$;

  ii. For $\mathbf{T} = \mathbf{V}(\mathbf{T}_\star)$, $\mathcal{G}[\mathbf{T}]$ is a c-component whose bidirected edges form a minimum spanning tree;

  iii. $\mathbf{T} = An(\mathbf{C})_{\mathcal{G}[\mathbf{T}]}$ for some $\mathbf{C}_\star \subseteq \mathbf{T}_\star$, with $\mathbf{C} = \mathbf{V}(\mathbf{C}_\star)$ and $\mathbf{c} = \mathbf{t} \cap \mathbf{C}_\star$;

  iv. Each vertex in $\mathcal{G}[\mathbf{T}]$ has at most one child; then

$\{\mathbf{T}_\star = \mathbf{t}\}$ is a *ctf-forest* rooted in $\{\mathbf{C}_\star = \mathbf{c}\}$. $\qquad\square$

**Definition 3.2** (Counterfactual (Ctf-) Hedge). Let $\{\mathbf{T}_\star = \mathbf{t}\}$ be a ctf-forest rooted in $\{\mathbf{C}_\star = \mathbf{c}\}$, having subgraph $\mathcal{G}$. If

  • $\mathbf{T}_\star \neq \mathbf{C}_\star$; and

---

[2] $\mathcal{A} = \varnothing$ corresponds to the observational distribution $P(\mathbf{v})$, while $\mathcal{A} = \{rand(\mathbf{X})\}$ is the standard randomized intervention on $\mathbf{X}$ and corresponds to the interventional distribution $P(\mathbf{v}; do(\mathbf{x}))$.

[3] We show in App. B.4 a more involved example where IDENTIFY[+] computes a ctf-factor using a sequence of non-trivial steps, beyond just marginalizing out extra terms.

---

**Algorithm 1** IDENTIFY[+]

1: **Input:** Causal diagram $\mathcal{G}$; ctf-factor $Q[\mathbf{C}_\star](\mathbf{c})$; ctf-factor $Q[\mathbf{T}_\star](\mathbf{t})$, s.t. $\mathbf{c}_\star \subseteq \mathbf{t}_\star$ and $\mathbf{V}(\mathbf{T}_\star)$ is a single c-component in $\mathcal{G}[\mathbf{V}(\mathbf{T}_\star)]$

2: **Output:** Expression for $Q[\mathbf{C}_\star](\mathbf{c}), \mathbf{c}_\star \subseteq \mathbf{t}_\star$, in terms of $Q[\mathbf{T}_\star](\mathbf{t})$; or **FAIL**

3: Let $\mathbf{H}_\star$ be the smallest set s.t. $\mathbf{C}_\star \subseteq \mathbf{H}_\star \subseteq \mathbf{T}_\star$ and there is no $C_{i_{[\mathbf{pa}_i]}} \in \mathbf{H}_\star, C_{j_{[\mathbf{pa}_j]}} \in \mathbf{T}_\star \setminus \mathbf{H}_\star$ where $\mathbf{t} \cap C_j \in \mathbf{pa}_i$

4: **if** $\mathbf{H}_\star = \mathbf{C}_\star$ **then**

5:     Return $Q[\mathbf{C}_\star](\mathbf{c}) = \sum_{\mathbf{t}\setminus\mathbf{c}} Q[\mathbf{T}_\star](\mathbf{t})$

6: **else if** $\mathbf{H}_\star = \mathbf{T}_\star$ **then**

7:     Return **FAIL**

8: **else if** $\mathbf{C}_\star \subset \mathbf{H}_\star \subset \mathbf{T}_\star$ **then**

9:     $Q[\mathbf{H}_\star](\mathbf{h}) = \sum_{\mathbf{t}\setminus\mathbf{h}} Q[\mathbf{T}_\star](\mathbf{t})$

10:     Let $\mathbf{H}_\star^1, ..., \mathbf{H}_\star^m$ be a partition of $\mathbf{H}_\star$ s.t. each $\mathbf{V}(\mathbf{H}_\star^i)$ forms a c-component in $\mathcal{G}[\mathbf{V}(\mathbf{H}_\star)]$

11:     Let $\mathbf{H}_\star^i$ be the subset s.t. $\mathbf{C}_\star \subseteq \mathbf{H}_\star^i$

12:     Compute $Q[\mathbf{H}_\star^i](\mathbf{h}^i)$ from $Q[\mathbf{H}_\star](\mathbf{h})$ by Thm. B.5

13:     Return IDENTIFY[+]$\left(\mathcal{G}, Q[\mathbf{C}_\star](\mathbf{c}), Q[\mathbf{H}_\star^i](\mathbf{h}^i)\right)$

14: **end if**

---

  • For each $V_{i_{[\mathbf{pa}_i]}} \in \mathbf{T}_\star \setminus \mathbf{C}_\star$ and $V_j \in Ch(V_i)_\mathcal{G}$, we have $\mathbf{t} \cap V_{i_{[\mathbf{pa}_i]}} = \mathbf{pa}_j \cap V_{i_{[\mathbf{pa}_i]}}$; that is, $\{\mathbf{t}_\star\}$ forms a "value chain" where each term's value is in its child's subscript for $\mathbf{T}_\star \setminus \mathbf{C}_\star$, then

$\{\mathbf{T}_\star = \mathbf{t}\}$ is a *counterfactual, or ctf-hedge* according to $\mathcal{G}$, rooted in $\{\mathbf{C}_\star = \mathbf{c}\}$. $\qquad\square$

Consider the minimum spanning tree in Fig. 3. $\{s, c, b_{s'c'}, d_{b'}, f_{d'}, e_{gh}\}$ is a ctf-forest rooted in $\{E_{gh} = e, F_{d'} = f\}$, while $\{s, c, b_{sc}, d_b, f_d, e_{gh}\}$ satisfies the definition of a ctf-hedge rooted in $\{E_{gh} = e, F_d = f\}$.

This structure marks an evolution of the previous hedge/thicket structures that have been used to witness non-identification (Shpitser & Pearl, 2006; Lee et al., 2019). This structure is designed to authenticate a failure to identify one ctf-factor from another, with a significantly simplified proof strategy as compared to Lee et al. (2019).

**Lemma 3.3** (Ctf-hedge non-identifiability). *Let $\{\mathbf{T}_\star = \mathbf{t}\}$ be a ctf-hedge rooted in $\{\mathbf{C}_\star = \mathbf{c}\}$, with subgraph $\mathcal{G}$. $Q[\mathbf{C}_\star](\mathbf{c})$ is not identifiable from $Q[\mathbf{T}_\star](\mathbf{t})$ given $\mathcal{G}$.* $\qquad\blacksquare$

**Lemma 3.4** (IDENTIFY[+] soundness and completeness). *Let $Q[\mathbf{T}_\star](\mathbf{t})$ be a ctf-factor in which each observable variable appears at most once, and $\mathcal{G}[\mathbf{V}(\mathbf{T}_\star)]$ is a c-component. Let $Q[\mathbf{C}_\star](\mathbf{c})$ be a ctf-factor s.t. $\mathbf{C}_\star \subseteq \mathbf{T}_\star, \mathbf{c} \subseteq \mathbf{t}$. $Q[\mathbf{C}_\star](\mathbf{c})$ is identifiable from $Q[\mathbf{T}_\star](\mathbf{t})$ and $\mathcal{G}$ iff IDENTIFY[+] returns an expression for it.*

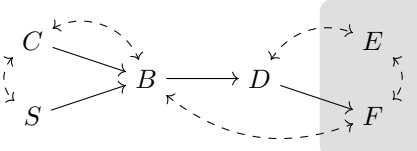

*Figure 3.* Subgraph of a ctf-hedge

*Proof sketch.* IDENTIFY$^+$ returns a valid expression, and only FAILS when it detects a ctf-hedge. ∎

This sub-routine forms a key component in our next contribution: the CTFIDU$^+$ algorithm (Alg. 2). CTFIDU$^+$ takes as input a graph $\mathcal{G}$, a counterfactual query $Q$, and a set of available distributions (including possibly counterfactual data) indexed by $\mathbb{A}$, and computes $Q$ iff it is identifiable from the input data. Naturally, CTFIDU$^+$ thus subsumes the previous identification algorithms in Table 1.

The algorithm works as follows: (a) we first remove redundant subscripts from the input query (Line 3-4); (b) we then expand the query into its ancestral set (Line 7); and (c) factorize this expression into smaller ctf-factors which are necessary and sufficient for identifying the query (Line 9); (d) we then process each input distribution (Line 11-14) and run the IDENTIFY$^+$ sub-routine using input ctf-factors (Line 16) to try and identify each target ctf-factor. If all the target terms are identified, these are combined into the final value (Line 22), or the algorithm FAILS (Line 20). We summarize these as Steps (i) to (viii) again in App. B.2. If the input query $P(\mathbf{Y}_\star = \mathbf{y})$ involves a nested counterfactual, previous work shows how to first convert the query into an equivalent summation of un-nested terms (Step 0-i. in Fig. 9) which can then individually be fed into Alg. 2.

**Theorem 3.5** (CTFIDU$^+$ soundness and completeness). *Given an un-nested counterfactual expression $\mathbf{Y}_\star$, $P(\mathbf{Y}_\star = \mathbf{y})$ is identifiable from a causal diagram $\mathcal{G}$ and a set of input distributions $\mathbb{A}$, iff CTFIDU$^+$ returns an expression for it.*

*Proof sketch.* An expression returned by CTFIDU$^+$ is valid. If CTFIDU$^+$ FAILS, this is because at least one of the necessary ctf-factors could not be identified from the input data. The failure of identification of this ctf-factor means the original query is non-identifiable from the input data. ∎

We show in App. B.5 an example of how CTFIDU$^+$ correctly retrieves the classic frontdoor-adjustment formula. We also show a more involved running example, where previous methods return FAIL. However, CTFIDU$^+$ recognizes the possibility of counterfactual data-collection, and returns an expression in terms of input data. Importantly, the input data is from a different regime than the query, and identification involves a sequence of non-trivial steps (Fig. 12).

---

**Algorithm 2** CTFIDU$^+$

1: **Input:** Causal diagram $\mathcal{G}$ over variables $\mathbf{V}$; un-nested counterfactual query $P(\mathbf{Y}_\star = \mathbf{y})$ involving variables in $\mathbf{V}$; input distribution specifications $\mathbb{A}$

2: **Output:** Expression for $P(\mathbf{Y}_\star = \mathbf{y})$, in terms of input distributions; or **FAIL** if not identifiable from $\langle \mathcal{G}, \mathbb{A} \rangle$

3: Let $\mathbf{Y}_\star \leftarrow ||\mathbf{Y}_\star||$, by Lem. B.3

4: **if** $\exists y_\mathbf{x}, y'_\mathbf{x} \in \mathbf{y}_\star$ or $y'_y \in \mathbf{y}_\star$, s.t. $y \neq y'$ **then**

5:     Return 0 (trivially impossible)

6: **end if**

7: Let $\mathbf{W}_\star = An(\mathbf{Y}_\star)$ (Def. B.2)

8: Let $P(\mathbf{W}_\star = \mathbf{w}) \leftarrow P(\mathbf{W}_\star = \mathbf{w})$ after applying the ancestral set transformation, or AST, to it (Thm. B.4)

9: Let $\mathbf{C}_\star^1, ..., \mathbf{C}_\star^k$ be a partition of $\mathbf{W}_\star$ s.t. each $\mathbf{V}(\mathbf{C}_\star^j)$ forms a c-component in $\mathcal{G}[\mathbf{V}(\mathbf{W}_\star)]$

10: **for** each $Q[\mathbf{C}_\star^j](\mathbf{c}^j)$ and $\mathcal{A} \in \mathbb{A}$ **do**

11:     $P(\mathbf{T}_\star = \mathbf{t}) \leftarrow$ REGIME-REGEX$(\mathcal{G}, \mathcal{A})$, Alg. 4

12:     Let $P(\mathbf{T}_\star = \mathbf{t}) \leftarrow P(\mathbf{T}_\star = \mathbf{t})$ after applying the ancestral set transformation to it (Thm. B.4)

13:     Let $\mathbf{T}_\star^1, ..., \mathbf{T}_\star^m$ be a partition of $\mathbf{T}_\star$ s.t. each $\mathbf{V}(\mathbf{T}_\star^i)$ forms a c-component in $\mathcal{G}$

14:     Compute each $Q[\mathbf{T}_\star^i](\mathbf{t}^i)$ from $P(\mathbf{T}_\star = \mathbf{t})$ using Thm. B.5

15:     **if** there exists some set $\mathbf{T}_\star^i$ s.t. $\mathbf{c}_\star^j \subseteq \mathbf{t}_\star^i$ and IDENTIFY$^+(\mathcal{G}, \mathbf{C}_\star^j, Q[\mathbf{T}_\star^i](\mathbf{t}^i))$ does not **FAIL** **then**

16:         $Q[\mathbf{C}_\star^j](\mathbf{c}^j) \leftarrow$ IDENTIFY$^+\Big(\mathcal{G}, Q[\mathbf{C}_\star^j], Q[\mathbf{T}_\star^i]\Big)$

17:     **end if**

18: **end for**

19: **if** some $Q[\mathbf{C}_\star^j](\mathbf{c}^j)$ was not identified from $\mathbb{A}$ **then**

20:     Return **FAIL**

21: **end if**

22: Return $P(\mathbf{Y}_\star = \mathbf{y}) \leftarrow \sum_{\mathbf{w} \setminus \mathbf{y}} \prod_j Q[\mathbf{C}_\star^j](\mathbf{c}^j)$

---

## 4. The Fundamental Limit of Identification

A natural follow-up question is how far up the PCH we can go using identification methods – are all $\mathcal{L}_3$ distributions now identifiable, in principle, when data is collected via *ctf-rand()*? Unfortunately, the answer is no. Next, we proceed to characterize the fundamental limit of exact causal inference from experimental data in the non-parametric setting.

Consider the graph in Fig. 5. Suppose we want to estimate a counterfactual quantity like $P(y_x \mid x')$. As discussed in Sec. 1, there are two approaches one could take. Counterfactual *identification* uses causal assumptions to reduce the query to a function of available data: for instance, $P(y_x \mid x') =$

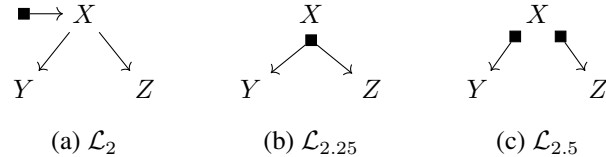

(a) $\mathcal{L}_2$        (b) $\mathcal{L}_{2.25}$        (c) $\mathcal{L}_{2.5}$

*Figure 4.* Difference in how an intervention on $X$ affects downstream variables in $\mathcal{L}_2$, $\mathcal{L}_{2.25}$, and $\mathcal{L}_{2.5}$.

$P(y; do(x))$. Counterfactual *realizability* involves directly sampling from the query's distribution via physical actions: for instance, drawing samples under *ctf-rand(X → A)* to get the distribution table of $P(x', a_x, y_x, z_x)$, from which we can directly retrieve the query.

Raghavan & Bareinboim (2025, Sec. 3) provided the first formal characterization of the family of $\mathcal{L}_3$ distributions which can be physically realized given the ability to perform actions like *rand()* and *ctf-rand()* on one or more variables. Notably, the authors showed that even if an environment permits maximal *ctf-rand()* interventions (which may not always be the case), not all distributions are realizable.

Yang & Bareinboim (2025, Defs. 11, 12) subsequently introduced a fine-grained segmentation of the PCH based on the experimenter's data-collection capabilities. Specifically, in addition to the familiar $\mathcal{L}_1, \mathcal{L}_2, \mathcal{L}_3$, they define *Layer 2.5* ($\mathcal{L}_{2.5}) \subseteq \mathcal{L}_3$) to delineate those counterfactual distributions which are realizable, in principle, if one were able to perform every possible *ctf-rand()* action. E.g., the distribution $P(y_x, z_{x'}, x'')$ w.r.t the graph in Fig. 4c can be realized by the two *ctf-rand()* interventions shown, and thus lies in $\mathcal{L}_{2.5}$. *Layer 2.25* ($\mathcal{L}_{2.25} \subseteq \mathcal{L}_{2.5}$) is a further refinement when *ctf-rand()* capabilities are more restricted and cannot be performed in a path-specific way, such as $P(y_x, z_x, x')$ as in Fig. 4b. In contrast, $\mathcal{L}_2$ involves erasing and replacing the natural value of the intervened variable via a standard *rand()* action, such as $P(y, z; do(x))$, shown in Fig. 4a.

Interestingly, whether a quantity falls within $\mathcal{L}_{2.5}$, i.e., whether its distribution can be realized via *ctf-rand()*, depends on the causal structure, and cannot always be determined from the form of the expression alone. For instance, given the graph in Fig. 5, the counterfactual $P(y_a \mid z_{a'}, a'')$ is realizable via *ctf-rand()* and lies within $\mathcal{L}_{2.5}$. But $P(y_x \mid z_{x'}, x'')$ is not physically realizable, and so lies beyond $\mathcal{L}_{2.5}$, that is, it belongs in $\mathcal{L}_3 \setminus \mathcal{L}_{2.5}$.[4]

We can now more formally rephrase the question with which we began this section: which $\mathcal{L}_3$ distributional quantities are identifiable in principle (for some graph $\mathcal{G}$), given access to some input data from $\mathcal{L}_{2.5}$? For instance, in Fig. 5, we

---

[4]Raghavan & Bareinboim (2025, Cor. 3.7) gives a simple graphical condition which detects whether a quantity lies in $\mathcal{L}_{2.5}$, which we reproduce in Thm. C.2 for ease of reference.

can show that the $\mathcal{L}_2$ quantity $P(y; do(x))$ is identifiable from the $\mathcal{L}_1$ distribution $P(x, y)$. The $\mathcal{L}_3$ quantity $P(z_a \mid a')$ is identifiable from the $\mathcal{L}_2$ distribution $P(z, a; do(x))$. What about the quantity $P(y_x|z_{x'}, x'')$, can it similarly be identified from some combination of counterfactual data? It turns out there are no identifiable quantities in $\mathcal{L}_3 \setminus \mathcal{L}_{2.5}$. I.e., *the limits of physical data-collection also impose a theoretical limit on which causal quantities can be point-identified in the non-parametric setting.*

**Theorem 4.1** (Limit of identification). *Given a query $Q$ belonging to $\mathcal{L}_i$ of the PCH and no lower layer, for every $j < i$ there exists a graph $\mathcal{G}$ s.t. $Q$ is identifiable from $\mathcal{G}$ and input data from $\mathcal{L}_j$, except for $i = 3$.* ∎

Perhaps surprisingly, this result means that there are $\mathcal{L}_2$ queries identifiable from $\mathcal{L}_1$ data, $\mathcal{L}_{2.25}$ queries identifiable from $\mathcal{L}_2$ data, and $\mathcal{L}_{2.5}$ queries identifiable from $\mathcal{L}_{2.25}$ data, but no purely-$\mathcal{L}_3$ queries identifiable from $\mathcal{L}_{2.5}$ data. E.g., in Fig. 5, $P(y_x|z_{x'}, x'')$ is fundamentally non-identifiable even from other realizable counterfactual data.

This barrier has considerable practical implications. E.g., take the $\mathcal{L}_3$ quantity known as the *natural total effect*, or NTE (Lee et al., 2025, Def. 2). While the details are out of scope, NTE is an important tool in the field of *explainable AI* (XAI). The **e**-specific NTE is defined as NTE($\mathbf{X}, Y \mid \mathbf{e}$) =

$$\mathbb{E}_{\mathbf{u} \sim P(\mathbf{U}|\mathbf{e}), \mathbf{u}' \sim P(\mathbf{U})}[Y_{\mathbf{X}(\mathbf{u})}(\mathbf{u}) - Y_{\mathbf{X}(\mathbf{u}')}(\mathbf{u})] \quad (7)$$

The first term $\mathbb{E}[Y_{\mathbf{X}(\mathbf{u})}(\mathbf{u})]$ works out to the expected observational outcome $\mathbb{E}[Y \mid \mathbf{e}]$. For a sub-population observed to have $\mathbf{E} = \mathbf{e}$ ($\mathbf{E} \subseteq \mathbf{V}$), the second term captures how the outcome would be affected if $\mathbf{X}$ were fixed by re-sampling from the observational distribution $P(\mathbf{X})$. This difference intuitively summarizes an explanation of how $\mathbf{X}$ affected an outcome $Y = y$ (see Bareinboim, 2025, Sec. 6.2.2.1).

For the example in Fig. 1, setting $\mathbf{X} = X$ and $\mathbf{e} = (x', y')$, the second term in Eq. 7 works out to

$$\mathbb{E}_{\mathbf{u} \sim P(\mathbf{U}|x', y'), \mathbf{u}' \sim P(\mathbf{U})}[Y_{X(\mathbf{u}')}(\mathbf{u})]$$
$$= \sum_{y, \mathbf{u}'} y . P(Y_{X(\mathbf{u}')} = y \mid x', y') P(\mathbf{u}') \quad (8)$$
$$= \sum_{y, x} y . P(y_x \mid x', y') P(x) \quad (9)$$

$P(y_x \mid x', y')$ is one of the famed *probabilities of causation* (Pearl, 1999). It can be shown that this quantity belongs to $\mathcal{L}_3 \setminus \mathcal{L}_{2.5}$. So, by Thm. 4.1, NTE cannot be identified, even with sophisticated counterfactual experimental capabilities – a relevant finding for the XAI community. Further, Thm. 4.1 points to a foundational connection between the seemingly orthogonal notions of counterfactual realizability and counterfactual identification.

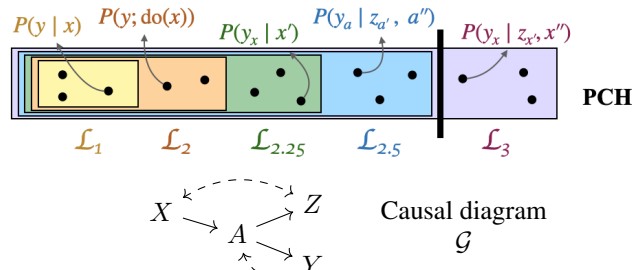

| Sample Query | Query Layer | Identifiable from | ID expression | |
|---|---|---|---|---|
| $P(y; \mathrm{do}(x))$ | $\mathcal{L}_2$ | $\mathcal{L}_1$ | $P(y \mid x)$ | ✔ |
| $P(y_x \mid x')$ | $\mathcal{L}_{2.25}$ | $\mathcal{L}_2$ | $P(y; \mathrm{do}(x))$ | ✔ |
| $P(y_a \mid z_{a'}, a'')$ | $\mathcal{L}_{2.5}$ | $\mathcal{L}_{2.25}$ | $P(y_a \mid a'')$ | ✔ |
| $P(y_x \mid z_{x'}, x'')$ | $\mathcal{L}_3$ | $\mathcal{L}_{2.5}$ | | ✘ |

*Figure 5.* $\mathcal{L}_{2.5}$ marks the theoretical limit of exact causal inference in the non-parametric setting (Thm. 4.1). Every layer of the PCH contains queries that may be identifiable using data from lower layers, except $\mathcal{L}_3 \setminus \mathcal{L}_{2.5}$.

**Corollary 4.2** (Id - realizability duality (informal)). *A query Q is identifiable from experimental and observational data and graph $\mathcal{G}$, if and only if it is realizable, in principle, using ctf-rand() actions.* ∎

The key insight of this duality is that *non-parametric identification of any causal quantity is essentially trying to mimic realizability*: any identifiable query should be answerable by sampling from a regime where we can, in principle, jointly observe each variable under randomized interventions of its parents (i.e., a *ctf-rand()* for every graph edge). This marks the limit of our data-collection capabilities, so if a query is still not realizable even in principle, such as $P(y_x \mid z_{x'}, x'')$ in Fig. 5, it means this query involves some confounding that cannot be disambiguated with any experimental data.

For the interested reader, we present in App. C.3 an intuition for this interplay using a *causal lattice* formulation. This perspective could inform future research into algorithm- and experiment-design for computing higher-order counterfactuals like NTE, such as by incorporating stronger assumptions to overcome bottlenecks along identification pathways.

## 5. Partial Identification using Ctf-Data

In this section, we show that even when a quantity is non-identifiable, the possibility of accessing counterfactual data through *ctf-rand()* can be used to derive provably tighter bounds for the *range* this quantity can take.

Sec. 4 discussed the task of *point identification*, and concluded that causal quantities beyond $\mathcal{L}_{2.5}$, such as the NTE, are non-identifiable from physically realizable data, in the non-parametric setting. Next, we discuss the *partial identification* of such causal quantities: given a causal diagram, we seek to use the available observational/experimental data to bound the range of possible values of this non-identifiable quantity. The bounds of this range are called *tight* if it is the smallest interval s.t. there exist SCMs where the causal quantity takes the boundary values (among the space of SCMs which satisfy the causal graph and the input data constraints). A range is *uninformative* if the bounds are $[0, 1]$. If a quantity is exactly identifiable, the tight range is simply

a point value, computable using the CTFIDU$^+$ algorithm.

Tian & Pearl (2000) first provided tight analytic bounds for non-identifiable counterfactuals known as the *probabilities of causation*, or PCs, which include quantities like $P(y_x \mid x', y')$ and $P(y_x, y'_{x'})$, assuming binary treatment. Subsequent work generalized this to bounds for PCs under non-binary treatments (Shu et al., 2026). These prior works all assume the input data is limited to observational or interventional distributions. It stands to reason that adding more input data using *ctf-rand()* can only tighten bounds further - if the constraint set is larger, the space of SCMs that satisfy it (and thus, the range of possible values of the identification query) is smaller.

**Proposition 5.1.** *Given causal diagram $\mathcal{G}$ and query $Q = P(\mathbf{y}_\star)$, let $[l, r]^{\mathbb{A}} \subseteq [0, 1]$ be the tight partial identification bounds for $Q$ given input data regimes $\mathbb{A}$. Then, for any $\mathbb{A}' \supset \mathbb{A}$, the bounds $[l, r]^{\mathbb{A}'} \subseteq [l, r]^{\mathbb{A}}$.* ∎

To make this concrete, consider the bow graph (Fig. 6.a) – a causal structure broadly representative of any real-world bivariate system where causation and unobserved confounding cannot be ruled out. For such environments, we next derive novel analytic bounds for NTE that are provably tighter than prior work, using realizable counterfactual data.

Specifically, suppose we want tight identification bounds for the $(x', y')$-specific NTE (Eq. 7). From Eq. 9, assuming that observational data $P(x)$ is already available, the bounds for NTE are determined by $P(y_x \mid x', y')$. Hence, we focus on deriving analytic bounds for this term.

If only observational data $P(\mathbf{V})$ is available, the bounds for $P(y_x \mid x', y')$ are uninformative, i.e., the range is the whole unit interval, as shown in yellow in Fig. 6.

**Lemma 5.2** (NTE - $\mathcal{L}_1$ bounds). *Given a bow graph causal structure (Fig. 6.a) and observational data $P(X, Y)$, the identification query $P(y_x \mid x', y'), x \neq x'$, is tightly bounded in the range $[0, 1]$.* ∎

If interventional data from a standard RCT is also available, this yields more informative and tighter bounds in terms of $P(y_x)$, depicted in orange in Fig. 6.

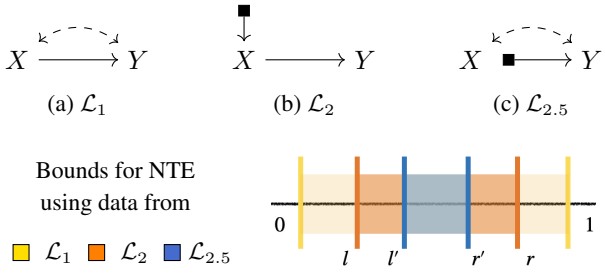

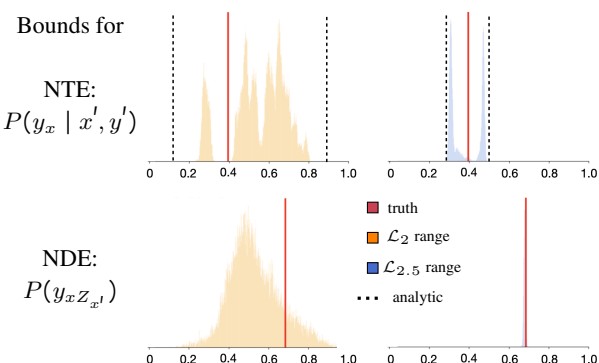

*Figure 6.* Increasingly tighter partial identification bounds for NTE using data from (a) $\mathcal{L}_1$, (b) $\mathcal{L}_2$, and (c) $\mathcal{L}_{2.5}$ regimes.

*Figure 7.* Example 2: partial identification bounds for NTE and NDE quantities. A density plot of values is generated by sampling from a Bayesian posterior over SCMs, given synthetic input data. The end-points of the range of values along the X-axis mark the empirically estimated range each quantity can take. Bounds are tighter when estimated using counterfactual data (blue) than interventional data (orange). Since NDE is exactly identifiable from counterfactual data, blue bounds collapse to the true value (red).

**Lemma 5.3** (NTE - $\mathcal{L}_2$ bounds). *Given a bow graph causal structure (Fig. 6.a), observational data $P(X, Y)$, and interventional data $P(Y_x)$, $\forall x$, the query $P(y_x \mid x', y'), x \neq x'$, is tightly bounded in the range $[l, r]$ defined as*

$$l = \max\left\{0, \frac{\alpha_{\min} - (1 - P(y' \mid x'))}{P(y' \mid x')}\right\} \quad (10)$$

$$r = \min\left\{1, \frac{\alpha_{\max}}{P(y' \mid x')}\right\}, \text{ where} \quad (11)$$

$$\alpha_{min} := \max\left\{0, \frac{P(y_x) - (1 - P(x'))}{P(x')}\right\} \quad (12)$$

$$\alpha_{max} := \min\left\{1, \frac{P(y_x)}{P(x')}\right\} \quad (13)$$

*Further,* $[l, r] \subseteq [0, 1]$ ∎

However, if the environment permits counterfactual data collection using *ctf-rand()*, this can be used to derive tighter bounds than the state of the art approach in Lem. 5.3, as shown in blue in Fig. 6. The novel bounds are as follows.

**Proposition 5.4** (NTE - $\mathcal{L}_{2.5}$ bounds). *Given a bow graph causal structure (Fig. 6.a), observational data $P(X, Y)$, interventional data $P(Y_x)$, and counterfactual data $P(Y_x \mid X)$, $\forall x$, the identification query $P(y_x \mid x', y'), x \neq x'$, is tightly bounded in the range $[l', r']$ defined as*

$$l' = \max\left\{0, \frac{P(y_x \mid x') - (1 - P(y' \mid x'))}{P(y' \mid x')}\right\} \quad (14)$$

$$r' = \min\left\{1, \frac{P(y_x \mid x')}{P(y' \mid x')}\right\} \quad (15)$$

*Further,* $[l', r'] \subseteq [l, r]$ *as defined in Lem. 5.3.* ∎

The new bounds are contained within the previous bounds when using only $\mathcal{L}_2$ data. The upshot of these results is that the standard approach in causal data science of using only observational and interventional data to bound important counterfactual quantities gives us loose bounds, which can be significantly improved if we could design experiments that permit counterfactual randomization. Future work could develop a more general framework to derive

tighter bounds for arbitrary counterfactual queries. As mentioned, the bounds for such counterfactuals are relevant in applications like algorithmic fairness and explainability.

Finally, we provide two examples illustrating how counterfactual data can, in practice, be used to empirically tighten bounds for identification queries. We use a Bayesian sampling methodology to estimate a $(1 - \beta)$ credible interval for the range of an identification query, and show that the range is tighter in practice when applying *ctf-rand()*.

**Example 2 (Traffic Camera - version 2).** Consider an expanded version of Example 1. Let $Y, Z \in \{0, 1\}$, $X \in \{0, 1, 2\}$. We now allow confounding between both $(X, Y)$ and $(Z, Y)$, to account for unlabeled driver tendencies affecting car color choice, or road obstructions affecting speeding, which may appear as video artifacts. We want to bound two identification queries: NDE $P(y_{xZ_{x'}})$ (Eq. 3), and NTE $P(y_x \mid x', y')$ (Eq. 9), which are useful for fairness analysis and explainability. We estimate bounds under two settings: (i) using data from $\mathcal{L}_1$ and $\mathcal{L}_2$ ; and (ii) using data from $\mathcal{L}_{2.5}$ obtained through *ctf-rand()* (Fig. 1c). We collect $N = 10^4$ samples per input distribution from a randomly generated (unobserved) SCM, and use this input data to empirically bound the queries by the methodology outlined earlier. The results are shown in Fig. 7.

For NTE, the 95% credible interval (*ci*) given $\mathcal{L}_{2.5}$ data (shown in blue) is significantly tighter than just using $\mathcal{L}_2$ data (in orange). Both empirical bounds validate the analytic bounds derived in Lem. 5.3 and Prop. 5.4, and contain the true NTE=0.395, indicated in red. For NDE, the 95% *ci* given $\mathcal{L}_2$ data (orange) is a range, but using $\mathcal{L}_{2.5}$ data causes the bounds (blue) to collapse to a single value – the true NDE=0.686 (in red). This validates the results in Sec. 3 and

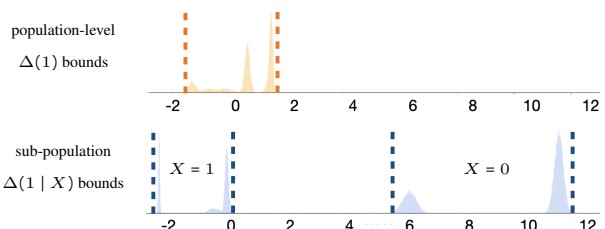

| Unit Type | $(Y_{X=0}, Y_{X=1})$ | Benefit: $\Delta(1)$ |
|-----------|----------------------|----------------------|
| Always-1 | (1,1) | $\gamma = 0$ |
| Helped | (0,1) | $\alpha = 20$ |
| Hurt | (1,0) | $\lambda = -5$ |
| Always-0 | (0,0) | $\delta = 0$ |

$\Delta(0) = 0$

*Figure 8.* Example 3: (Left) benefit function by unit type; (Centre) estimated bounds of population-level benefit using $\mathcal{L}_2$ data (orange) and sub-population level benefit using $\mathcal{L}_{2.5}$ data (blue); (Right) counterfactual strategy dominates standard interventional approach.

the CTFIDU$^+$ algorithm since, as derived in Eq. 3, NDE is indeed identifiable using counterfactual randomization. These findings are consistent across five random SCM specifications. Details of the sampling process, and the random SCMs are provided in App. D.1.

We replicate these findings using two public datasets: the *Oregon Health Insurance* dataset and *Project STAR* dataset. In both examples, bounds computed using counterfactual data are significantly narrower than using $\mathcal{L}_2$ data alone. Details are provided in App. D.3.

**Example 3 (Unit Selection, Li & Pearl (2019)).** Consider a drug rehab program with the causal diagram in Fig. 6a. $X$ indicates treatment assignment. $Y$ indicates treatment success. Each participant is of a *canonical type* (Angrist et al., 1996; Balke & Pearl, 1997) defined in Fig. 8(left). E.g., the *Helped* type of participant ($Y_{X=0} = 0, Y_{X=1} = 1$) would recover iff offered treatment, while the *Always-0* type of participant ($Y_{X=0} = 0, Y_{X=1} = 0$) would not recover whether they received treatment or not. A given participant's type and the population probability of each type are unknown. A *unit selection* problem assigns a benefit $\Delta(1 \mid \text{type})$ for each unit type receiving treatment, as $\gamma, \alpha, \lambda, \delta$, respectively. The baseline benefit of non-treatment $\Delta(0 \mid \text{type}) = 0$, for all types. We wish to maximize avg. treatment benefit, given input data from observations/experiments.[5] We evaluate two strategies: (1) the standard approach in Li & Pearl (2022) of using $\mathcal{L}_2$ data to empirically bound the quantity $P(y'_{X=0}, y_{X=1}), \forall y, y'$, then combining these to bound $\Delta(1)$ to make a population-level treatment decision; (2) a counterfactual decision strategy (Bareinboim et al., 2015; Raghavan & Bareinboim, 2025) using *ctf-rand()* to collect $\mathcal{L}_{2.5}$ data, then using this to bound $P(y'_{X=0}, y_{X=1}|x')$, and thus estimate the $\Delta(1|X)$ to make a *subpopulation*-level decision for $X = x'$.

The results are in Fig. 8(center, right). Since $\mathcal{L}_{2.5}$ data allows more granularity, strategy (2) counterintuitively assigns treatment $do(X = 1)$ only to units who would have naturally received $X = 0$, as their benefit range is entirely positive. Strategy (1) is strictly suboptimal as it either as-

signs 0 to everyone, or forces 1 even on units with natural $X = 1$, for whom the benefit range is entirely negative. Further details of input data and the counterfactual strategy are provided in App. D.2.

## 6. Conclusion

We developed the CTFIDU$^+$ algorithm (Alg. 2), a complete method for identifying counterfactuals given an arbitrary collection of physically realizable input data (Thm. 3.5). Previous completeness results for counterfactual identification assumed that input data is restricted to $\mathcal{L}_1$ and $\mathcal{L}_2$ of the PCH. We then proved the theoretical limit to exact nonparametric identification, demonstrating a foundational duality between counterfactual identifiability and realizability (Thm. 4.1, Cor. 4.2). Finally, even if a counterfactual quantity is non-identifiable, we demonstrate that counterfactual data still remains practically valuable by yielding tighter empirical bounds for the range of the quantity.

**Limitations.** (1) The first obvious limitation is that we assume knowledge of a causal graph. This is a standard assumption needed to make progress in many areas of causal inference. Future work could incorporate uncertainty over graph misspecification; (2) Second, our work is about deriving an *identification formula* in terms of input data. We are agnostic to the subsequent pipeline used to *estimate* this formula using finite samples. Depending on the estimation method chosen, this step may be sample inefficient or exhibit unstable convergence; (3) Third, our work assumes discrete variables. In principle, Alg. 2 can handle jointly-continuous variables by swapping PMFs with PDFs. In practice, some steps may become computationally and statistically challenging, and workarounds like discretization may disrupt the downstream data-analysis; (4) Lastly, we are addressing a setting where it is possible to gather counterfactual data using *ctf-rand()*, which may not always be feasible. However, we argue this is increasingly relevant in generative AI pipelines or automated business workflows, which allow more controllability and granularity of interventions. Prior work points out how these results can stimulate novel experiment design ideas in this new area of causal inference (Raghavan & Bareinboim, 2025).

---

[5]As a special case, if $\gamma = \delta = 0$ and $\lambda = -\alpha$, this works out to maximizing the *avg. treatment effect* (ATE) of $X$ on $Y$.

# Acknowledgements

This research is supported in part by the NSF, ONR, AFOSR, DoE, Amazon, JP Morgan, and The Alfred P. Sloan Foundation.

# Impact Statement

This paper presents work whose goal is to advance the fields of machine learning and causal reasoning. Our results help researchers better understand the limits of what can be inferred from observational and experimental data. There are many potential societal consequences of our work, none which we feel must be specifically highlighted here.

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

# Appendix Contents

# A. Graphical terminology

Structural Causal Models (SCM) and causal diagrams are described in the preliminaries in Sec. 1. See Bareinboim et al. (2022) for full treatment. We use the following graphical kinship nomenclature w.r.t causal diagram $\mathcal{G}$:

- Parents of $V_i$, denoted $\mathbf{Pa}_i$: the set $\{V_j\}$ s.t. there is a direct edge $V_j \to V_i$ in $\mathcal{G}$. $\mathbf{Pa}_i$ does not include $V_i$.

- Children of $V_i$, denoted $\mathbf{Ch}(V_i)$: the set $\{V_j\}$ s.t. there is a direct edge $V_i \to V_j$ in $\mathcal{G}$. $\mathbf{Ch}(V_i)$ does not include $V_i$.

- Ancestors of $V_i$, denoted $\mathbf{An}(V_i)$: the set $\{V_j\}$ s.t. there is a path (possibly length 0) from $V_j$ to $V_i$ consisting only of edges pointing toward $V_i$, $V_j \to ... \to V_i$. $\mathbf{An}(V_i)$ is defined to include $V_i$.

- Descendants of $V_i$, denoted $\mathbf{Desc}(V_i)$: the set $\{V_j\}$ s.t. there is a path (possibly length 0) from $V_i$ to $V_j$ consisting only of edges pointing toward $V_j$, $V_i \to ... \to V_j$. $\mathbf{Desc}(V_i)$ is defined to include $V_i$.

- Non-descendants of $V_i$, denoted $\mathbf{NDesc}(V_i)$: the set $\mathbf{V} \setminus \mathbf{Desc}(V_i)$. $\mathbf{NDesc}(V_i)$ does not include $V_i$.

Given a graph $\mathcal{G}$, $\mathcal{G}_{\overline{\mathbf{X}}\underline{\mathbf{W}}}$ is the result of removing edges coming into variables in $\mathbf{X}$, and edges coming out of $\mathbf{W}$.

$\mathcal{G}[\mathbf{W}]$ denotes a vertex-induced subgraph, which includes only $\mathbf{W}$ and the edges among its elements. $\mathcal{G}[\mathbf{V}(\mathbf{W}_\star)]$ denotes the subgraph which includes only the vertices $\{V \mid V_\mathbf{x} \in \mathbf{W}_\star\}$ and the edges among its elements.

# B. Tools for Counterfactual Identification

In this appendix, we summarize all the components used in the task of counterfactual identification, including a review of prior work. Sec. B.1 can be skipped or skimmed if readers are already familiar with these results.

## B.1. Previous results

Below are some relevant conceptual components developed in prior work (Correa et al., 2021; Correa & Bareinboim, 2025).

Given an arbitrarily nested counterfactual expression, the Counterfactual Un-nesting Theorem, or CUT, provides a way to compute it in terms of un-nested probability terms.

**Theorem B.1** (Counterfactual Un-nesting Theorem (CUT)). *Let $Y, X \in \mathbf{V}, \mathbf{T}, \mathbf{Z} \subseteq \mathbf{V}$, and let $\mathbf{z}$ be a set of values for $\mathbf{Z}$. Then, the nested counterfactual $P(Y_{T_\star X_z} = y)$ can be written as an un-nested counterfactual, as follows:*

$$P(Y_{T_\star X_z} = y) = \sum_x P(Y_{T_\star x} = y, X_z = x), \tag{16}$$

*where the subscript $\star$ is a wildcard for an arbitrarily nested counterfactual clause.* ∎

This can be recursively applied to get fully un-nested terms. For instance, for the diagram in Fig. 1, we can write $P(y_{xZ_{x'}}) = \sum_z P(y_{zw}, z_{x'})$.

**Definition B.2** (Ancestors of a counterfactual). Given a causal diagram $\mathcal{G}$ and a potential response $Y_\mathbf{x}$, the set of *(counterfactual) ancestors* of $Y_\mathbf{x}$, denoted $An(Y_\mathbf{x})$, consists of each $W_\mathbf{z}$ s.t. $W \in An(Y)_{\mathcal{G}_\underline{\mathbf{X}}}$, and $\mathbf{z} = \mathbf{x} \cap An(W)_{\mathcal{G}_\overline{\mathbf{X}}}$. For a set $\mathbf{W}_\star$, $An(\mathbf{W}_\star)$ is defined to be the union of the ancestors of each potential response in the set. ∎

This generalizes the notion of ancestors of a causal variable to the ancestors of potential responses under different regimes. For instance, for the diagram in Fig. 1, $An(Y_x) = \{Y_x, Z_x\}$ and $An(Y_z) = \{Y_z, X\}$.

**Lemma B.3** (Exclusion operator). *The exclusion operator $||.||$ when applied to a potential response returns the minimal counterfactual subscript set, removing redundant interventions (e.g. non-ancestors) from the subscript.*

*Consider a causal diagram $\mathcal{G}$ and a potential response $Y_\mathbf{x}$. Let $||Y_\mathbf{x}|| := Y_\mathbf{z}$, where $\mathbf{Z} = \mathbf{X} \cap An(Y)_{\mathcal{G}_\overline{\mathbf{X}}}$ and $\mathbf{z} = \mathbf{x} \cap \mathbf{Z}$.*

*Then, $||Y_\mathbf{x}|| = Y_\mathbf{x}$ holds for any model compatible with $\mathcal{G}$.* ∎

If a counterfactual expression is ancestral (i.e. it contains its own ancestors), the following results shows how to convert it into a ctf-factor expression, and further decompose it based on c-components.

**Theorem B.4** (Ancestral Set Transformation (AST)). *Let* $\mathbf{W}_\star$ *be an ancestral set, that is,* $An(\mathbf{W}_\star) = \mathbf{W}_\star$, *and let* $\mathbf{w}$ *be a vector with the values of each variable in* $\mathbf{W}_\star$. *Then,* $P(\mathbf{W}_\star = \mathbf{w})$ *can be rewritten in ctf-factor format as follows,*

$$P(\mathbf{W}_\star = \mathbf{w}) = P(\bigwedge_{W_\mathbf{t} \in \mathbf{W}_\star} W_{\mathbf{pa}_W} = w), \tag{17}$$

*where each* $w$ *is* $w_\mathbf{t}$ *and* $\mathbf{pa}_w$ *is determined for each* $W_\mathbf{t} \in \mathbf{W}_\star$ *as follows:*

*(i) the values for variables in* $\mathbf{Pa}_w \cap \mathbf{T}$ *are the same as in* $\mathbf{t}$, *and*

*(ii) the values for variables in* $\mathbf{Pa}_w \setminus \mathbf{T}$ *are taken from* $\mathbf{w}$ *corresponding to the parents of* $\mathbf{W}$. ∎

**Theorem B.5** (Counterfactual factorization). *Let* $Q[\mathbf{H}_\star](\mathbf{h}) = P(\mathbf{H}_\star = \mathbf{h})$ *be a ctf-factor. Let* $\mathbf{H}^1, ..., \mathbf{H}^m$ *be the c-components in* $\mathcal{G}[\mathbf{V}(\mathbf{H}_\star)]$. *Define* $\mathbf{H}^i_\star = \{H_{\mathbf{pa}_h} \in \mathbf{H}_\star \mid H \in \mathbf{H}^i\}$ *and* $\mathbf{h}^i$ *as the values in* $\mathbf{h}$ *corresponding to* $\mathbf{H}^i_\star$. *Note that* $\mathbf{H}^1_\star, ..., \mathbf{H}^m_\star$ *form a partition of* $\mathbf{H}_\star$. *Then, we have that* $Q[\mathbf{H}_\star](\mathbf{h})$ *decomposes as*

$$Q[\mathbf{H}_\star](\mathbf{h}) = P(\mathbf{H}_\star = \mathbf{h}) = \prod_i P(\mathbf{H}^i_\star = \mathbf{h}^i) \tag{18}$$

*Furthermore, let* $H_1 < H_2 < ...$ *be a topological order over the variables in* $\mathcal{G}[\mathbf{V}(\mathbf{H}_\star)]$. *Then, each factor can be computed from* $P(\mathbf{H}_\star = \mathbf{h})$ *as*

$$Q[\mathbf{H}^i_\star](\mathbf{h}^i) = P(\mathbf{H}^i_\star = \mathbf{h}^i) = \prod_{H_j \in \mathbf{H}^i} \frac{\sum_{\{h \mid H_{\mathbf{pa}_h} \in \mathbf{H}_\star, H_j < H\}} P(\mathbf{H}_\star = \mathbf{h})}{\sum_{\{h \mid H_{\mathbf{pa}_h} \in \mathbf{H}_\star, H_{j-1} < H\}} P(\mathbf{H}_\star = \mathbf{h})} \tag{19}$$

∎

Next, we discuss how to classify a ctf-factor as "consistent", based on conflicts in the counterfactual terms. And how to convert a consistent ctf-factor into a Layer 2 c-factor.

**Definition B.6** (Consistent ctf-factor). A ctf-factor $Q[\mathbf{C}_\star](\mathbf{c}) = P(\mathbf{C}_\star = \mathbf{c})$ is called *consistent* if it does not contain two counterfactuals $X_{\mathbf{pa}_x}, Y_{\mathbf{pa}_y} \in \mathbf{C}_\star$ with values $x, y$ such that any pair of values in $x \cup y \cup \mathbf{pa}_x \cup \mathbf{pa}_y$ conflict. Otherwise, the ctf-factor is called *inconsistent*. ∎

**Lemma B.7** (Collapsing operation). *If a ctf-factor* $Q[\mathbf{C}_\star](\mathbf{c})$ *is consistent, then it is equivalent to the Layer 2 confounded (c-) factor, as follows,*

$$Q[\mathbf{C}_\star](\mathbf{c}) = Q[\mathbf{C}](\mathbf{v}), \text{ with } \mathbf{v} \text{ consistent with } \mathbf{c} \text{ and the subscripts in } \mathbf{C}_\star, \tag{20}$$

*where the c-factor is defined in Eq. 6.* ∎

Finally, we reproduce for ease of reference the classic IDENTIFY algorithm that provides a method to identify a Layer 2 c-factor from an input c-factor.

---

**Algorithm 3** IDENTIFY (Tian & Pearl, 2003, Sec. 4.4)

---

1: **Input:** Causal diagram $\mathcal{G}$; set $\mathbf{C} \subseteq \mathbf{T} \subseteq \mathbf{V}$ s.t. $\mathcal{G}[\mathbf{T}]$ has one single c-component; c-factor $Q[\mathbf{T}](\mathbf{v})$

2: **Output:** Expression for $Q[\mathbf{C}](\mathbf{v})$ in terms of $Q[\mathbf{T}](\mathbf{v})$; or **FAIL**

3: Let $\mathbf{H} := An(\mathbf{C})$ in $\mathcal{G}_\mathbf{T}$

4: **if** $\mathbf{H} = \mathbf{C}$ **then**

5:     Return $Q[\mathbf{C}](\mathbf{v}) = \sum_{\mathbf{t} \setminus \mathbf{c}} Q[\mathbf{T}](\mathbf{v})$

6: **else if** $\mathbf{H} = \mathbf{T}$ **then**

7:     Return **FAIL**

8: **else if** $\mathbf{C} \subset \mathbf{H} \subset \mathbf{T}$ **then**

9:     $Q[\mathbf{H}](\mathbf{v}) = \sum_{\mathbf{t} \setminus \mathbf{h}} Q[\mathbf{T}](\mathbf{v})$

10:     Let $\mathbf{H}^i$ be the ctf c-component in $\mathbf{H}$ according to $\mathcal{G}_\mathbf{H}$ s.t. $\mathbf{C} \subseteq \mathbf{H}^i$

11:     Compute $Q[\mathbf{H}^i](\mathbf{v})$ from $Q[\mathbf{H}](\mathbf{v})$ using Theorem B.8

12:     Return IDENTIFY$(\mathcal{G}, \mathbf{C}, Q[\mathbf{H}^i](\mathbf{v}))$

13: **end if**

---

**Theorem B.8** (C-factor decomposition (Lem. 4, ibid.))**.** *Given* $\mathbf{H} \subseteq \mathbf{V}$*, let* $\mathbf{H}$ *be partitioned into c-components* $\mathbf{H}^1, ..., \mathbf{H}^m$ *in the subgraph* $\mathcal{G}_{\mathbf{H}}$*. Then,* $Q[\mathbf{H}](\mathbf{v})$ *decomposes as*

$$Q[\mathbf{H}](\mathbf{v}) = \prod_i Q[\mathbf{H}^i](\mathbf{v}) \tag{21}$$

*Furthermore, let* $H_1 < H_2 < ...$ *be a topological ordering of the variables in* $\mathcal{G}[\mathbf{H}]$*. Let* $\mathbf{H}^{(\leq j)} := \{H_1, ..., H_j\}$ *be the set of variables in* $\mathbf{H}$ *ordered up to and including* $H_j$*, with* $\mathbf{H}^{(\leq 0)} := \varnothing$*. Then, each* $Q[\mathbf{H}^i]$ *is computable from* $Q[\mathbf{H}](\mathbf{v})$ *and given by*

$$Q[\mathbf{H}^i] = \prod_{H_j \in \mathbf{H}^i} \frac{Q[\mathbf{H}^{(\leq j)}]}{Q[\mathbf{H}^{(\leq j-1)}]}, \tag{22}$$

*where each* $Q[\mathbf{H}^{(\leq j)}](\mathbf{v})$ *can be computed simply as*

$$Q[\mathbf{H}^{(\leq j)}] = \sum_{\mathbf{h} \backslash \mathbf{h}^{(\leq j)}} Q[\mathbf{H}] \tag{23}$$

*The vector* $(\mathbf{v})$ *is omitted from Eqs. 22,23 for legibility.* ∎

### B.2. Steps to identification

We summarize in Fig. 9 the steps involved in algorithmic identification from counterfactual (Layer 3) data.

As a pre-processing step, if the query is a nested counterfactual $P(\mathbf{Y}'_\star = \mathbf{y}')$,

Step 0: Map it to un-nested terms, $P(\mathbf{Y}_\star = \mathbf{y})$ using the Un-Nesting Theorem (Thm. B.1) which is now the input to the CTFIDU$^+$ algorithm.

Steps i-v of CTFIDU$^+$ (Alg. 2) involve

Steps ii: Remove any redundant subscripts (such as interventions on non-ancestors) or trivial counterfactuals (Lines 3-6)

Step iii-iv: Expand the query into the set of its counterfactual ancestors (Def. B.2) (Line 7); identifying this expression is both necessary and sufficient to identify the query; re-write this expression in ctf-factor format (Line 8);

Steps v: Factorize this ctf-factor into smaller ctf-factors based on the confounding structure in the graph, using the ctf-factorization formulas (Thm. B.5); identifying each of these smaller ctf-factors is both necessary and sufficient to identify the query (Line 9).

These steps are the same as the prior work which designed the CTFID algorithm (Correa et al., 2021, Alg. 1) for counterfactual identification from Layer 2 data, we merely extend the proof of necessity of Step iii. for Layer 3 input data.

After this stage, the prior CTFID maps each of these ctf-factor terms to Layer 2 c-factors *only if* it is "consistent", and then applies the celebrated IDENTIFY algorithm to identify each of these c-factors from input Layer 2 data (**gray-dotted** box, Steps vi-x in Fig. 9). If all ctf-factors in Step v. are thus identified, these terms can be chained to compute the query. Otherwise, identification **FAILS**.

By contrast, Steps vi-viii of the new CTFIDU$^+$ algorithm (red boxes in Fig. 9) involve:

Steps vi: For each input data regime $\mathcal{A}$, pre-process using the helper function and AST Thm. to generate the input data expression in ctf-factor format (Line 11-12);

Steps vii: Map this to the smaller ctf-factors we can compute from this data, based on confounding structure (Line 13-14). These ctf-factors are sufficient to identify the query, if it is identifiable from the input data;

Steps viii: For each query ctf-factor, find an input ctf-factor that contains it, and run the novel IDENTIFY$^+$ algorithm to identify the query term from the input term (Line 16);

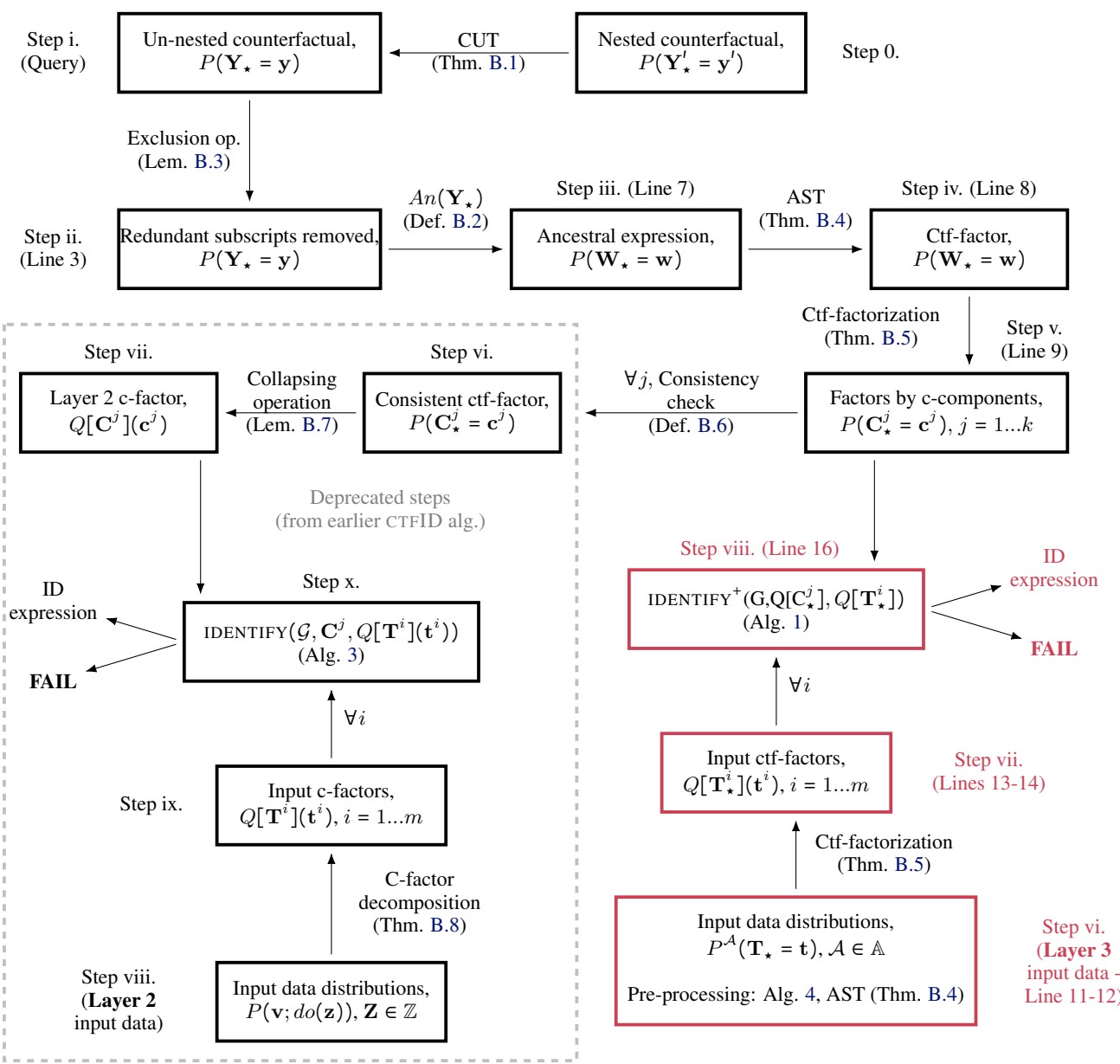

*Figure 9.* Algorithmic steps for counterfactual identification. Steps in the **Gray-dotted** box illustrate the deprecated steps of the old CTFID algorithm from prior work (Correa et al., 2021), which only allows identification from Layer 2 input data. **Red** boxes illustrate the new CTFIDU$^+$ algorithm (Alg. 2), which allows identification from Layer 3 input data. Steps 0-v are shared by both.

If all query ctf-factors from Step v. are identified in Step viii, these terms can be chained to identify the query (Line 22). Otherwise, identification **FAILS**. The necessity and sufficiency of each step (in particular, the IDENTIFY$^+$ subroutine) provide the proof for the soundness and completeness of the CTFIDU$^+$ algorithm (Thm. 3.5).

## B.3. Complexity of CTFIDU$^+$

It was shown by Correa & Bareinboim (2025) that constructing an ancestor set for $\mathbf{Y}_\star$ is $O(z(n+m))$, where $n, m, z$, and $d$ refer to the number of nodes, edges, (different) interventions in $\mathbf{Y}_\star$, and maximum cardinality of any observable variable in $\mathcal{G}$, respectively. Since a realizable input distribution has at most $n$ terms, IDENTIFY$^+$ can be invoked up to $O(zn(n+m))$

times in the main loop. And each inner-loop can be invoked up to $n$ times. The time complexity is $O(zn^2(n+m))$.

## B.4. Example using IDENTIFY$^+$

Below, we show an example of using the IDENTIFY$^+$ sub-routine to compute a ctf-factor if and only if it is identifiable from another ctf-factor. Each step of the computation is non-trivial, and cannot be skipped by just marginalizing out extra terms.

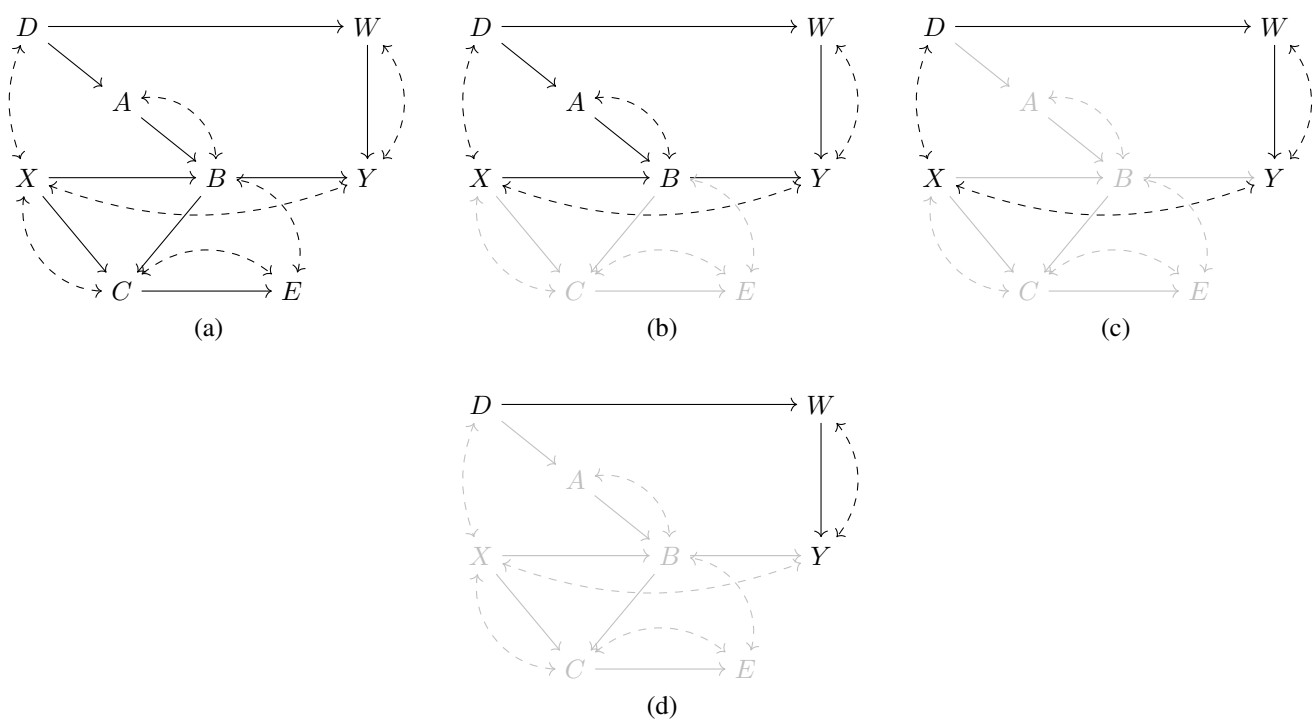

(a)         (b)         (c)

(d)

*Figure 10.* Example of using IDENTIFY$^+$ to identify a ctf-factor from an input ctf-factor. Each step is a recursive call to IDENTIFY$^+$.

**Example B.9.** *Given the graph in Fig. 10(a), consider the sub-routine call of* IDENTIFY$^+$ *where we want*

**Target ctf-factor**, $Q[\mathbf{C}_\star](\mathbf{c}) = P(w'_d, y_{b'w})$

**Input ctf-factor**, $Q[\mathbf{T}_\star](\mathbf{t}) = P(d, a_d, x, b'_{ax}, c_{bx'}, e_c, w'_d, y_{b'w})$ *computable from the input data.*

**Function call**: IDENTIFY$^+\left(\mathcal{G}, P(w'_d, y_{b'w}), P(d, a_d, x, b'_{ax}, c_{bx'}, e_c, w'_d, y_{b'w})\right)$ *goes through the following steps:*

1. *Line 3 :* $\mathbf{H}_\star \triangleq P(d, a_d, x, b'_{ax}, w'_d, y_{b'w})$

   - *This is the minimal subset* $\mathbf{h}_\star \supseteq \mathbf{c}_\star$ *without any subscripts appearing in the values-vector of* $\mathbf{t}_\star \setminus \mathbf{h}_\star$. *Because* $d$ *appears in the subscript of* $a_d$, *and* $a$ *appears in the subscript of* $b'_{ax}$ *etc., we can't shift any more terms from* $\mathbf{h}_\star$ *to* $\mathbf{t}_\star \setminus \mathbf{h}_\star$. *Note: subscripts are value sensitive and* $b \neq b'$, *for instance.*

2. *Line 9 :* $Q[\mathbf{H}_\star](\mathbf{h}) = P(d, a_d, x, b'_{ax}, w'_d, y_{b'w}) = \sum_{c,e} P(\mathbf{T}_\star = \mathbf{t})$

   - *Marginalize out* $\{C, E\}$, *giving us the graph in Fig. 10b.*

   - *Probability axioms don't permit marginalizing out any more terms.*

3. *Line 10 : the c-components in the subgraph in Fig. 10b are* $\{D, X, W, Y\}$ *and* $\{A, B\}$

4. *Line 11 :* $\mathbf{H}^i_\star \triangleq P(d, x, w'_d, y_{b'w})$

   - *This corresponds to the smallest c-component containing* $\mathbf{C}_\star$, *inducing the subgraph in Fig. 10c.*

5. *Line 12 : Compute $Q[\mathbf{H}_\star^i](\mathbf{h}^i) = P(d, x, w_d', y_{b'w})$ from $Q[\mathbf{H}_\star](\mathbf{h})$ using the ctf-factorization theorem (Thm. B.5)*

6. *Line 13 : **Function call** $\text{IDENTIFY}^+\left(\mathcal{G}, P(w_d', y_{b'w}), P(d, x, w_d', y_{b'w})\right)$*

7. *Line 3 : $\mathbf{H}_\star \triangleq P(d, w_d', y_{b'w})$*

   - *This is the minimal subset $\mathbf{h}_\star \supseteq \mathbf{c}_\star$ without any subscripts appearing in the values-vector of $\mathbf{h}_\star'$. Because $d$ appears in the subscript of $w_d'$, we can't shift any more terms from $\mathbf{h}_\star$ to $\mathbf{h}_\star'$*

8. *Line 9 : $Q[\mathbf{H}_\star](\mathbf{h}) = P(d, w_d', y_{b'w}) = \sum_x P(\mathbf{T}_\star = \mathbf{t})$*

   - *Marginalize out $\{X\}$, giving us the graph in Fig. 10d.*

   - *Probability axioms don't permit marginalizing out any more terms.*

9. *Line 10 : the c-components in the subgraph in Fig. 10d are $\{D\}$ and $\{W, Y\}$*

10. *Line 11 : $\mathbf{H}_\star^i \triangleq P(w_d', y_{b'w})$*

   - *This corresponds to the smallest c-component containing $\mathbf{C}_\star$.*

11. *Line 12 : Compute $Q[\mathbf{H}_\star^i](\mathbf{h}^i) = P(w_d', y_{b'w})$ from $Q[\mathbf{H}_\star](\mathbf{h})$ using the ctf-factorization theorem (Thm. B.5)*

12. *Line 13 : **Function call** $\text{IDENTIFY}^+\left(\mathcal{G}, P(w_d', y_{b'w}), P(w_d', y_{b'w})\right)$ immediately returns $P(w_d', y_{b'w})$ as needed.*

## B.5. Examples using CTFIDU$^+$

Below, we show two examples of a counterfactual query given a causal graph, and how the CTFIDU$^+$ algorithm identifies this query from counterfactual data. For a breakdown of the steps involved, refer to Sec. B.2.

**Example B.10** (Front-Door). *Consider the front-door graph shown in Fig. 11. We show how the CTFIDU$^+$ algorithm correctly retrieves the front-door adjustment formula. Our query is $P(y; do(x)) = P(y_x)$ and input is the observational distribution $P(x', z, y)$. The query chain in Steps iii-v decomposes the query in the ctf-factors that we need to identify: $P(z_x), P(y_z)$.*

*The input distribution is then rewritten in ctf-factor format (Step vi) and decomposed into constituent ctf-factors by c-components which can be computed from the input data using Thm. B.5 to get $P(x', y_z) = P(y \mid z, x').P(x')$ and $P(z_x) = P(z \mid x)$. $\text{IDENTIFY}^+(\mathcal{G}, P(y_z), P(x', y_z))$ immediately returns $P(y_z) = \sum_{x'} P(x, y_z)$. Composing these and marginalizing as the final step in Line 22, we get $P(y_x) = \sum_z P(z_x, y_x) = \sum_z P(z \mid x) \sum_{x'} P(y \mid z, x')P(x')$.*

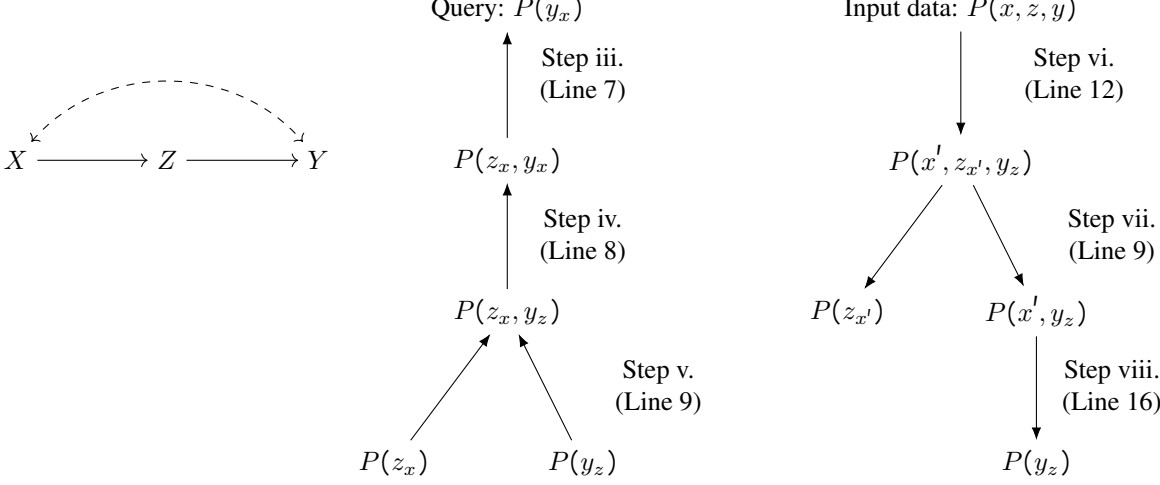

*Figure 11.* Example B.10 showing how the CTFIDU$^+$ algorithm (Alg. 2) correctly retrieves the front-door adjustment formula.

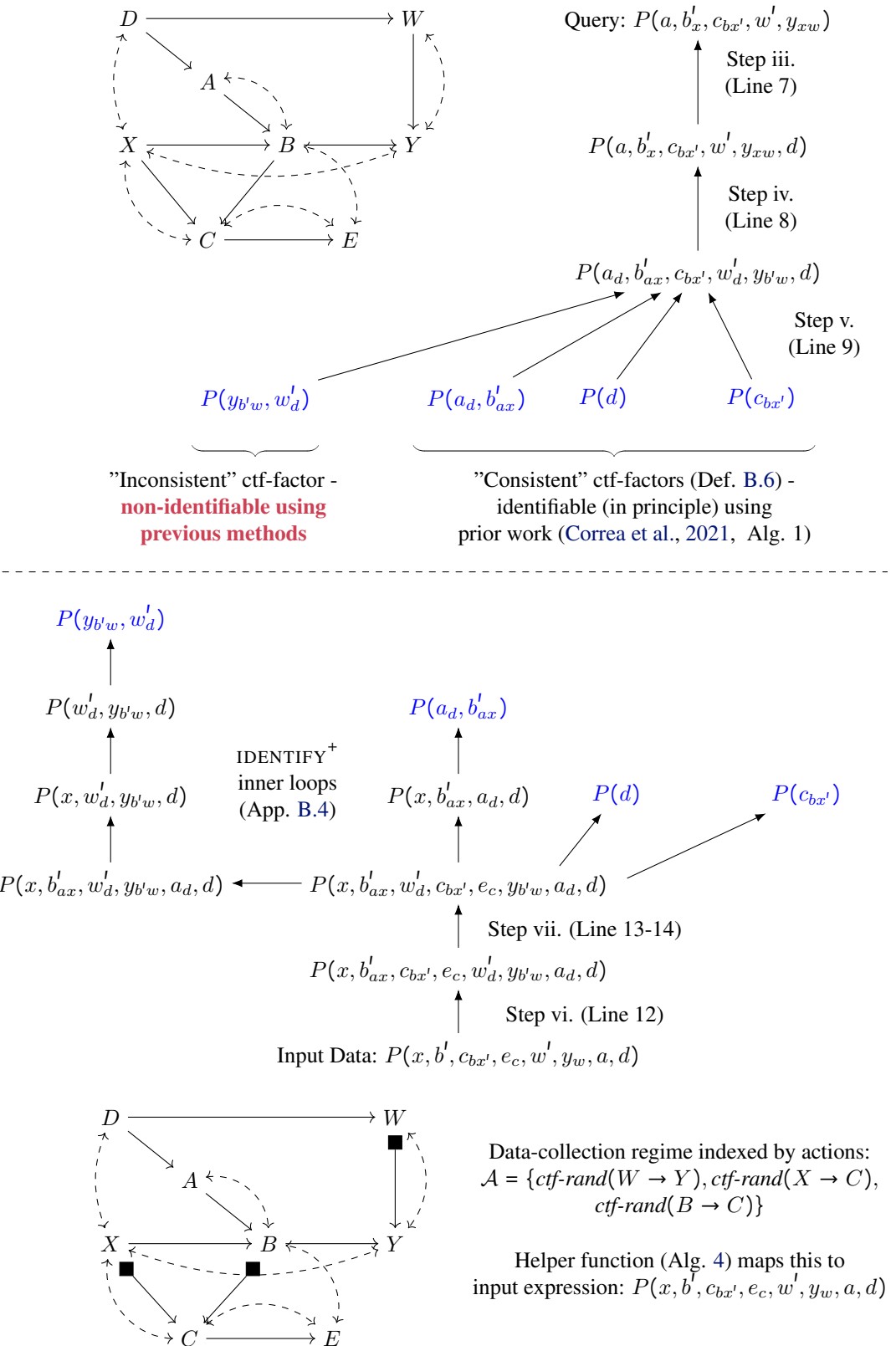

*Figure 12.* Example B.11 involving a causal diagram (top left) and Layer 3 query (top right). The new CTFIDU⁺ algorithm (Alg. 2) follows the steps shown, reaching the four ctf-factors that are both necessary and sufficient to identify, in order to identify the root query. Next, counterfactual data from an experimental regime (bottom) is used to systematically identify each of these ctf-factors.

**Example B.11.** *Consider the graph $\mathcal{G}$ and the query $Q = P(a, b'_x, c_{bx'}, w', y_{xw})$ shown in Fig. 12 (top left, top right).*

*Calling Alg. 2 as* $\textsc{ctfidu}^+(\mathcal{G}, Q, \mathcal{A})$ *follows the steps shown in the tree sequence in Fig. 12 (upper half). First, it expands the query into an ancestral expression, and then rewrites it in ctf-factor format.*

*Next, it decomposes this expression into the four smaller ctf-factors that are both necessary and sufficient to identify, in order to identify the root query. These are the four blue "leaf" terms in Fig. 12 (upper half). One of these terms, $P(y_{b'w}, w'_d)$, is "inconsistent" per Def. B.6, and therefore non-identifiable using the previous* $\textsc{ctfidu}$ *algorithm (Correa et al., 2021, Alg. 1). At this point, previous methods will return* **FAIL**, *because they assume the input data is only from Layer 2.*

*However, it is possible to gather Layer 3 data through counterfactual randomization, as discussed in Sec. 1. Fig. 12 (bottom half) illustrates data collection under the actions $\mathcal{A} = \{ctf\text{-}rand(W \to Y), ctf\text{-}rand(X \to C), ctf\text{-}rand(B \to C)\}$. Mapping this to an input expression, we see that the input distribution is not trivially identical to the query expression we began with.*

$\textsc{ctfidu}^+$ *proceeds to rewrite this input distribution in ctf-factor format $Q[\mathbf{T_\star}]$. There is no further decomposition at this stage, since all the variables belong to one c-component. This ctf-factor is then used to identify each of the leaf nodes in Fig. 12 (upper half) by calling* $\textsc{identify}^+(\mathcal{G}, Q[\mathbf{C_\star}], Q[\mathbf{T_\star}])$ *for each leaf node $\mathbf{C_\star}$ in turn. Two of these leaf nodes are immediately computable by a simple marginalization step. The remaining two are identified by following non-trivial inner loops.* □

## B.6. Sanity checks for completeness

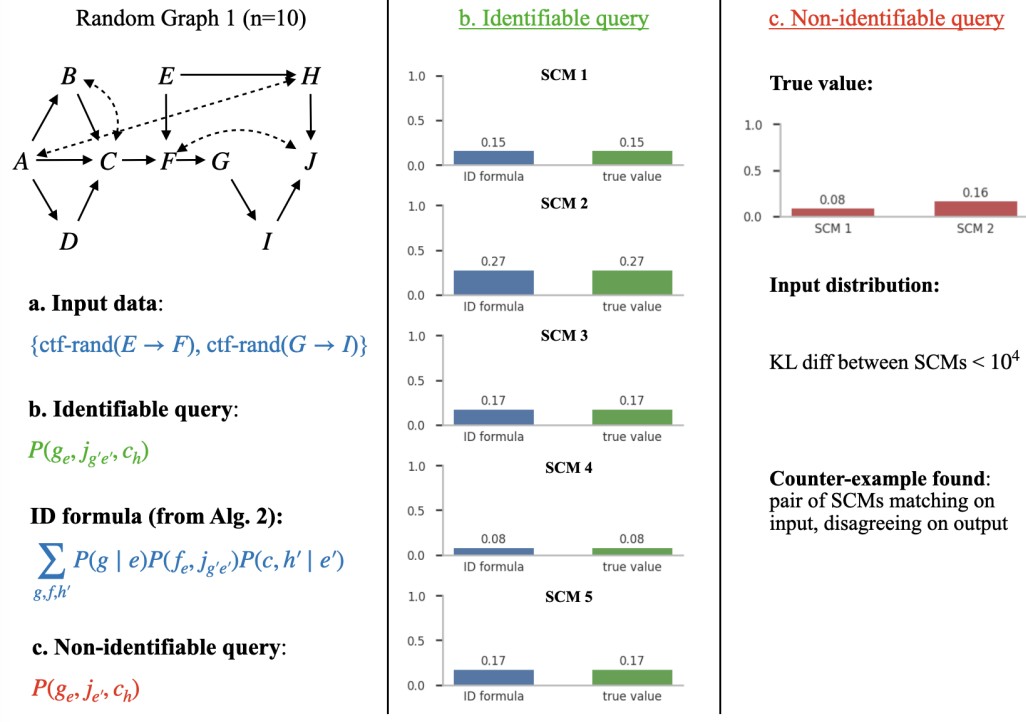

*Figure 13.* Sanity check for the completeness of Alg. 2 using a randomly generated graph. The ID formula gives the correct answer for the counterfactual query in multiple random SCMs compatible with the graph. The non-identifiable query indeed elicits a counterexample of SCMs witnessing non-ID.

We demonstrate the following sanity check for algorithmic completeness of Alg. 2. By design, we need to use random simulated SCMs because the main point of "completeness" is that *for all possible underlying causal models compatible with the given causal graph, the identification formula gives the correct answer*; furthermore, where Alg. 2 FAILS, we can guarantee non-identifiability. This shows that our method works for any valid ground-truth SCM.

The pipeline is quite straightforward:

- We generate a random causal graph having 3-10 variables, with arbitrary edges, arbitrary confounding structure.

- For the graph, we manually specify (a) some input $\mathcal{L}_{2.5}$ data; (b) a query that is identifiable from this data, with the identification formula given by Alg. 2; (c) a query that is not identifiable from this data.

- For the identifiable query in (b), we generate 5 random causal models and show that in each model, the identification formula given by Alg. 2 (blue column in Fig. 13) exactly matches the true value (green column in figures); these can be computed exactly from the SCM.

- For the non-identifiable query in (c), through rejection-sampling we generate 2 causal models that agree on input and disagree on output distributions (a standard way of proving that a query is non-identifiable).

We performed this sanity check using 5 randomly generated graphs, and show the results for one of the graphs in Fig. 13. Thus, we show that our identification method is sound and complete.

## C. Limits of Identification and Realizability

In this appendix, we review the definitions of Layers 2.25 and 2.5. We also provide a helpful intuition for framing the results in Sec. 4. We follow the terminology in Raghavan & Bareinboim (2025); Yang & Bareinboim (2025).

### C.1. Layer 2.5 ($\mathcal{L}_{2.5}$)

Given a causal diagram $\mathcal{G}$, Yang & Bareinboim (2025) define Layer 2.5 ($\mathcal{L}_{2.5}$) of the PCH to contain precisely those counterfactual distributions from which it is hypothetically possible to draw samples, if the environment permitted this *ctf-rand()* procedure for all variables. Below is the formal definition, followed by an intuitive explanation.

**Definition C.1** (Layer 2.5). An SCM $\mathcal{M} = \langle \mathbf{U}, \mathbf{V}, \mathcal{F}, P(\mathbf{u}) \rangle$ induces a family of joint distributions over $\mathbf{V}$, indexed by each interventional variable set $\mathbf{X}$. Layer 2.5, or $\mathcal{L}_{2.5}$ of the PCH is defined to contain all distributions satisfying the following expression. For $\mathbf{X}, \mathbf{Y} \subseteq \mathbf{V}$:

$$P^{\mathcal{M}}\left( \bigwedge_{V_i \in \mathbf{Y} \setminus \mathbf{X}} V_{i_{[\mathbf{x}_i]}} = v_i, \bigwedge_{V_i \in \mathbf{Y} \cap \mathbf{X}, v_i = V_i \cap \mathbf{x}} V_{i_{[\mathbf{x}_i \setminus v_i]}} = v_i \right)$$

$$= \sum_{\mathbf{u}} \mathbb{1}\left[ \bigwedge_{V_i \in \mathbf{Y} \setminus \mathbf{X}} V_{i_{[\mathbf{x}_i]}}(\mathbf{u}) = v_i, \bigwedge_{V_i \in \mathbf{Y} \cap \mathbf{X}, v_i = V_i \cap \mathbf{x}} V_{i_{[\mathbf{x}_i \setminus v_i]}}(\mathbf{u}) = v_i \right] P(\mathbf{u}), \text{ where} \tag{24}$$

i. the variables in the subscript for each term, $\mathbf{X}_i \subseteq \mathbf{X}$, $\mathbf{x}_i \in Domain(\mathbf{X}_i)$, and $\bigcup_i \mathbf{X}_i = \mathbf{X}$; and

ii. for any $V_i$ and any $B \in \mathbf{X} \cap \mathbf{Pa}_i$, and for all $V_j \in \mathbf{Y}$: if $V_i \notin \mathbf{X}_j$ and $V_i \in \mathbf{An}(V_j)$ in $\mathcal{M}_{\mathbf{x}}$, then $\mathbf{x}_i \cap B = \mathbf{x}_j \cap B$. ∎

For example, in Fig. 14a, we see how one can draw samples directly from the distribution $P(y_x, z_{x'})$ by performing *ctf-rand(X → Y)* and *ctf-rand(X → Z)* separately. However, this distribution is not physically *realizable* if the graph were per Fig. 14b - the mediator $A$ is a bottleneck and can only receive one value, either $x$ or $x'$. The causal structure matters. In Fig. 14, given graph $\mathcal{G}_1$, $P(y_x, z_{x'})$ is an $\mathcal{L}_{2.5}$ distribution. But given $\mathcal{G}_2$, $P(y_x, z_{x'})$ lies outside $\mathcal{L}_{2.5}$.

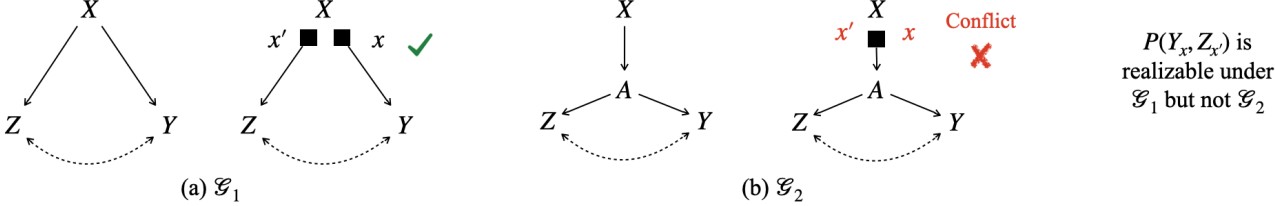

*Figure 14.* It is physically possible to draw samples from $P(y_x, z_{x'})$ given graph $\mathcal{G}_1$ using *ctf-rand()*, but not given $\mathcal{G}_2$.

**Theorem C.2.** *(Raghavan & Bareinboim, 2025, Cor. 3.7) Given causal diagram $\mathcal{G}$, a counterfactual distribution $P(\mathbf{Y}_\star)$ belongs to Layer 2.5 (i.e., it is physically realizable, in principle) iff the counterfactual ancestor set $An(\mathbf{Y}_\star)$ (Def. B.2) does not contain a pair of potential responses $W_{\mathbf{t}}, W_{\mathbf{s}}$ of the same variable $W$ under different regimes $\mathbf{t} \neq \mathbf{s}$.* ∎

A simple way to test if some $P(\mathbf{Y}_\star)$ belongs to $\mathcal{L}_{2.5}$ is to list the counterfactual ancestors (Def. B.2) of $\mathbf{Y}_\star$. Given $\mathcal{G}_2$ in Fig. 14, $An(Y_x, Z_{x'}) = \{Y_x, Z_{x'}, A_x, A_{x'}\}$ which contains both $A_x, A_{x'}$ and thus $P(y_x, z_{x'})$ lies outside $\mathcal{L}_{2.5}$.

### C.2. Layer 2.25 ($\mathcal{L}_{2.25}$)

$\mathcal{L}_{2.5}$ contains distributions which are realizable possibly using multiple *ctf-rand()* procedures for the same variable, such as $P(y_x, z_{x'})$ in Fig. 14a. Layer 2.25 ($\mathcal{L}_{2.25}$) is a subset of $\mathcal{L}_{2.5}$, containing only the distributions which can be realized using at most one *ctf-rand()* procedure per variable.

**Definition C.3** (Layer 2.25). As SCM $\mathcal{M} = \langle \mathbf{U}, \mathbf{V}, \mathcal{F}, P(\mathbf{u}) \rangle$ induces a family of joint distributions over $\mathbf{V}$, indexed by each interventional value set $\mathbf{x}$. Layer 2.25, or $\mathcal{L}_{2.25}$ of the PCH is defined to contain all distributions satisfying the following expression. For $\mathbf{X}, \mathbf{Y} \subseteq \mathbf{V}$ and $\mathbf{x} \in Domain(\mathbf{X})$:

$$P^{\mathcal{M}} \left( \bigwedge_{V_i \in \mathbf{Y} \setminus \mathbf{X}} V_{i_{[\mathbf{x}_i]}} = v_i, \bigwedge_{V_i \in \mathbf{Y} \cap \mathbf{X}, v_i = V_i \cap \mathbf{x}} V_{i_{[\mathbf{x}_i \setminus v_i]}} = v_i \right)$$

$$= \sum_{\mathbf{u}} \mathbb{1}\left[ \bigwedge_{V_i \in \mathbf{Y} \setminus \mathbf{X}} V_{i_{[\mathbf{x}_i]}}(\mathbf{u}) = v_i, \bigwedge_{V_i \in \mathbf{Y} \cap \mathbf{X}, v_i = V_i \cap \mathbf{x}} V_{i_{[\mathbf{x}_i \setminus v_i]}}(\mathbf{u}) = v_i \right] P(\mathbf{u}), \text{ where} \tag{25}$$

i.  the interventional subscript for each term, $\mathbf{x}_i \subseteq \mathbf{x}$ and $\bigcup_i \mathbf{x}_i = \mathbf{x}$; and

ii. for any $v_i \in \mathbf{x}$ and all $V_j \in \mathbf{Y}$, if $V_i \in An(V_j)$ in $\mathcal{M}_{\mathbf{x} \setminus V_j}$, then $v_j \in \mathbf{x}_j$. ∎

A visual intuition for these layers is provided in the examples in Fig. 15.

a.  $\mathcal{L}_1$ (Fig. 15a) simply represents the observational regime of the system under its natural behavior.

b.  $\mathcal{L}_2$ (Fig. 15b) represents interventional regimes, where a standard randomization action *rand(X)* is used to override and fix some variable $X$ in the system.

c.  $\mathcal{L}_{2.25}$ (Fig. 15c) represents counterfactual distributions which can be physically realized using counterfactual randomization actions of the form *ctf-rand(X → $\mathbf{Ch}(X)$)*, where at most one randomization is permitted per variable in a way that affects all outgoing causal paths from the variable.

d.  $\mathcal{L}_{2.5}$ (Fig. 15d) generalizes this to all counterfactual distributions which can be physically realized using multiple *ctf-rand(X → C)* actions per variable, in a way that may affect separate downstream variables differently.

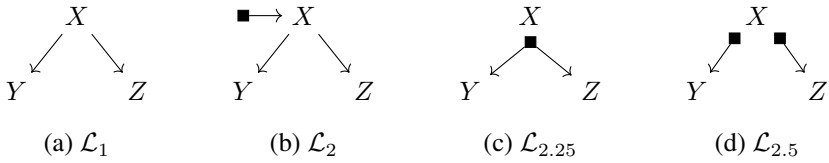

| (a) $\mathcal{L}_1$ | (b) $\mathcal{L}_2$ | (c) $\mathcal{L}_{2.25}$ | (d) $\mathcal{L}_{2.5}$ |

*Figure 15.* Difference in how an intervention on $X$ affects downstream variables in $\mathcal{L}_1$, $\mathcal{L}_2$, $\mathcal{L}_{2.25}$, and $\mathcal{L}_{2.5}$.

### C.3. Causal lattice framework

We describe in this subsection an intuition for Thm. 4.1 and Cor. 4.2 using a causal lattice over the *ctf-factors* that can be generated from an input distribution (see Sec. 2 for a definition of a *ctf-factor*). This lattice functions as an **inference generator**: all the nodes in the lattice are effectively causal quantities that can be identified using input data distributions.

We formulate our ***causal lattice*** as follows. Given a causal diagram and input data distributions:

- The ***source nodes*** of the causal lattice are all the available (i.e. input) data distributions.

- Each node has ***outgoing edges*** to all the distributional quantities that can be computed using the former node. Outgoing edges include

  – Mapping from a source node to an equivalent *ctf-factor* formulation (Thm. B.4): 1-to-1 connection

  – Decomposing a larger *ctf-factor* into smaller *ctf-factors* (Eq. 19): 1-to-many connections

  – Composing smaller *ctf-factors* into a larger *ctf-factor* (Eq. 18): many-to-1 connections

  – Mapping from a *ctf-factor* to an equivalent non-*ctf-factor* distribution, if the latter is ancestral (Thm. B.4): 1-to-1 connection

  – Marginalization of a distribution to get a smaller distribution: 1-to-1 connection

- There could be multiple valid pathways from the set of input data distributions to a particular quantity of interest, via different sets of intermediate nodes.

**Example C.4.** *In Fig. 12, conjoining the respective distribution trees in the upper and lower half of the figure would constitute a valid sub-lattice of the causal lattice induced by the input data distribution $P(x, b', c_{bx'}, w', y_w, a, d)$.*  □

Next, we define a way to rank the level of "inconsistency" that characterizes any given *ctf-factor*. This could be seen as a generalization of Def. B.6 from Correa et al. (2021).

**Definition C.5** (Ctf-factor inconsistency level)**.** A ctf-factor is said to have an inconsistency level, as defined by the table in Fig. 16. If the ctf-factor satisfies several rows, the highest number is chosen (see Sec. 2 for a definition of a ctf-factor). ■

| Inconsistency Level | Definition: ctf-factor contains… | Examples | | Causal diagram $\mathcal{G}$ |
|---|---|---|---|---|
| 5 | $y_{\mathbf{pa}_Y}, y'_{\mathbf{pa}'_Y}$ s.t. $\mathbf{pa}_Y \neq \mathbf{pa}'_Y$ | $P(y_x, y_{x'})$ | ■■■■■ | |
| 4 | $y_{\mathbf{pa}_Y}, x_{\mathbf{pa}_X}$ s.t. $\mathbf{pa}_Y$ and $\mathbf{pa}_X$ disagree on $\mathbf{Pa}_Y \cap \mathbf{Pa}_X$ | $P(y_x, z_{x'})$ | ■■■■□ | |
| 3 | $y_{\mathbf{pa}_Y}, x_{\mathbf{pa}_X}$ s.t. $\mathbf{pa}_Y$ and $x$ disagree on $\mathbf{Pa}_Y \cap X$ | $P(y_x, x')$ | ■■■□□ | |
| 2 | $y_{\mathbf{pa}_Y}$ and $\exists X \in \mathbf{V}$ s.t. $(X, Y)$ share a bidirected edge in $\mathcal{G}$, but ctf-factor does not contain any $x_{\mathbf{pa}_X}$ | $P(x)$ , $P(y_x, z_x)$ | ■■□□□ | |
| 1 | none of the above inconsistencies | $P(y_x, x, z_x)$ | ■□□□□ | |

*Figure 16.* Levels of inconsistency of a ctf-factor, with examples for the causal diagram shown on the right.

We illustrate this in the example provided in Fig. 17, which shows sections of the causal lattice generated from an example causal graph and available input distributions. Each node is a distribution which can be identified from the input data, culminating in all possible target quantities which are identifiable. Nodes which are *ctf-factors* are colored **blue**, and assigned an inconsistency level per Fig. 16.

The key insights of this causal lattice presentation are as follows:

- Input distribution nodes belonging to $\mathcal{L}_1, \mathcal{L}_2, \mathcal{L}_{2.25}, \mathcal{L}_{2.5}$ (respectively) have outgoing edges to *ctf-factors* of $\leq$ inconsistency level 1, 2, 3, 4 (respectively). E.g. the observational distribution $P(w, x, a, z, y)$ can only point to *ctf-factors* of inconsistency level 1, shown on the left in Fig. 17.

- Different types of arrows from some ctf-factor(s) to other(s) can change the inconsistency levels differently:

  – A 1-to-many arrow can decrease inconsistency level from the preceding node. E.g., in the section marked (ii) in Fig. 17, inconsistency level goes from 3 to 1.

  – A 1-to-1 arrow can reduce inconsistency level, or can increase it from 1 to 2. E.g., in the section marked (i) in Fig. 17, inconsistency level goes from 1 to 2.

  – A many-to-1 arrow can increase inconsistency level over each of the preceding nodes. E.g., in the section marked (iii) in Fig. 17, inconsistency level goes from 3,1,1 to 4.

- Target output nodes belonging to $\mathcal{L}_1, \mathcal{L}_2, \mathcal{L}_{2.25}, \mathcal{L}_{2.5}$ (respectively) have incoming edges from *ctf-factors* of $\leq$ inconsistency level 1, 2, 3, 4 (respectively). E.g. the interventional distribution $P(w_x, a_x)$ requires an incoming edge from *ctf-factor* of inconsistency level 2, shown on the left in Fig. 17.

- $\mathcal{L}_3$ output nodes have incoming edges from other $\mathcal{L}_3$ nodes or *ctf-factors* of inconsistency level 5. And each *ctf-factor* of inconsistency level 5 has incoming edges from other $\mathcal{L}_3$ nodes or *ctf-factors* of inconsistency level 5.

**This increase/decrease in inconsistency along lattice pathways is what allows higher-order counterfactual quantities from $\mathcal{L}_i$ to be identified from lower-layer $\mathcal{L}_j$ data, $j < i$.**

However, the last point is a fundamental limitation. If we want an $\mathcal{L}_3$ output node, it needs an incoming edge from a node with inconsistency level 5. The reasoning is provided in the proof of Thm. 4.1.

**By induction, it follows that no quantity in $\mathcal{L}_3 \setminus \mathcal{L}_{2.5}$ is identifiable because there is no lattice path to it starting from a physically realizable input data distribution** (as illustration on the bottom right in Fig. 17).

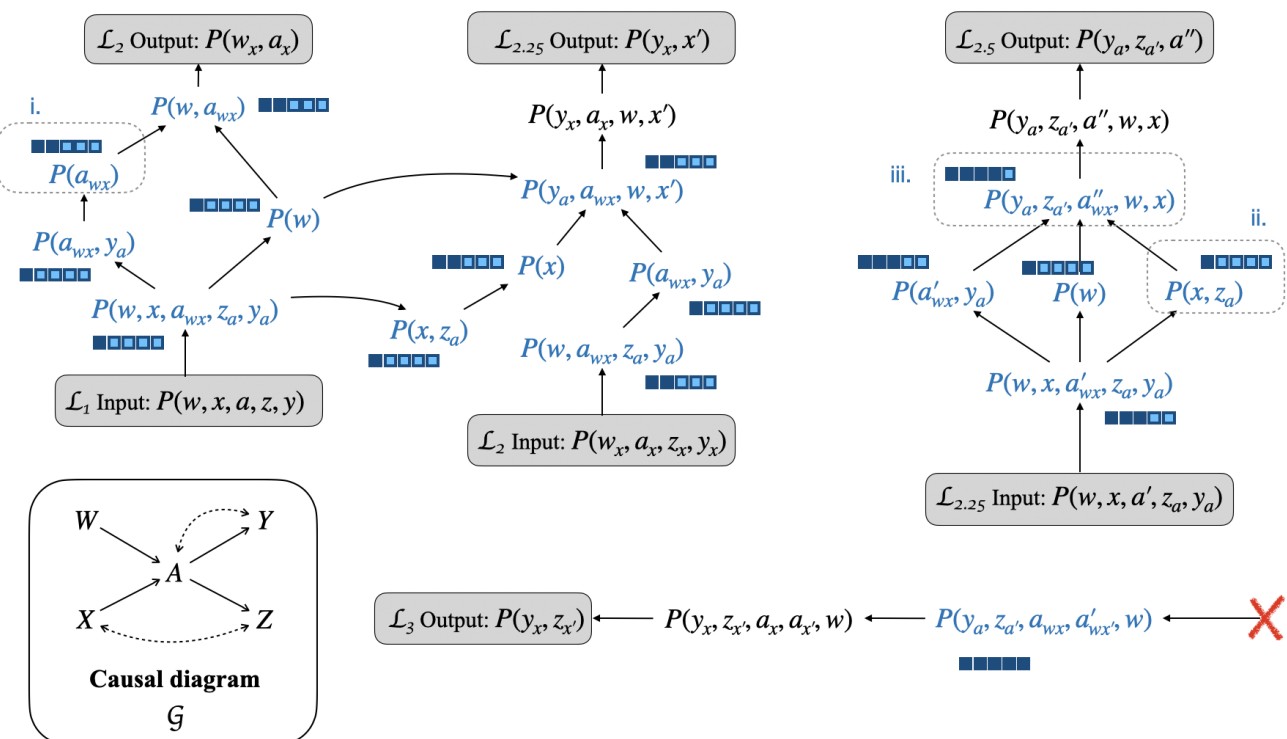

*Figure 17.* Example illustrating sections of the causal lattice generated from a graph $\mathcal{G}$, and input data distributions as source nodes. Each subsequent node is a distribution which can be identified from all distributions pointing into it. **Blue** nodes are *ctf-factors*, each assigned an inconsistency level per Fig. 16. $\mathcal{L}_3$ output nodes don't have a valid lattice path starting from a realizable input data distribution.

# D. Partial Identification: Example Details

The simulation code is provided in the supplementary material, for reproducibility. We follow a *Markov Chain Monte Carlo* (MCMC) methodology developed in Zhang et al. (2022) to derive empirical bounds for quantities of interest:

We generate synthetic input datasets from a random underlying (hidden) SCM. With this data, we derive a posterior distribution over all possible SCMs compatible with the causal graph and input data. Sampling from this posterior, we get a distribution over the values for our target query, giving us a range of feasible values. We repeat this with 5 random SCMs to ensure consistent results.

Hyperparameters: $N = 10^4$ samples per input distribution; credible interval 95%.

### D.1. Example 2

The causal graph for this example is shown in Fig. 18a.

Causal assumptions: $Y$ represents an automated AI decision to issue a speeding ticket to a driver based on video footage. $X$ represents the color of the driver's car. $Z$ is an indicator of whether the driver was over the speed limit or not. $X$ might affect $Z$ if pedestrians and other drivers react to, say, a red car and affect its speeding. $X$ might affect $Y$ directly due to a high correlation in training data between the color preference of different socioeconomic groups and their speeding tendency. Speeding and outcome might be affected by an unobserved confounder - unlabeled road obstacles (which present as video artifacts). Car color and outcome might be affected by an unobserved confounder - unlabeled driver attributes (which can be picked up in video footage).

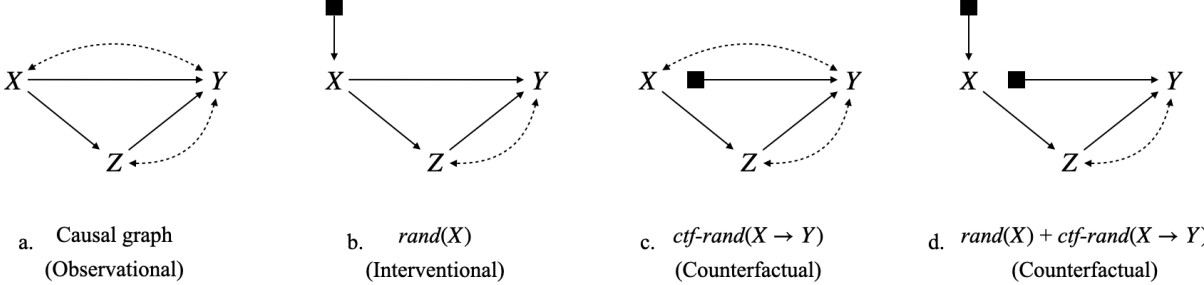

|  |  |  |  |
|---|---|---|---|
| a. Causal graph (Observational) | b. *rand*($X$) (Interventional) | c. *ctf-rand*($X \to Y$) (Counterfactual) | d. *rand*($X$) + *ctf-rand*($X \to Y$) (Counterfactual) |

*Figure 18.* Causal diagram and different data-collection regimes for Example 2 (Traffic Camera v2).

We are interested in obtaining empirical bounds for two queries:

(i) NTE-like $P(Y_{X=1}|X=0, Y=0)$: comparison between using observational + interventional input data (orange plots) vs. using counterfactual input data from the regime shown in Fig. 18c (blue plots)

(ii) NDE-like $P(Y_{X=1,Z_{X=0}}=1)$: comparison between using observational + interventional input data (orange plots) vs. using counterfactual input data from the regime shown in Fig. 18d (blue plots)

Results: in Fig. 19 we show results for each query across 5 randomly generated underlying true causal models. Across all examples, using counterfactual data (blue plots) narrows the credible interval for the query vs using observational and/or interventional data alone (orange plots). The true target value is indicated by a red line.

### D.2. Example 3

The causal graph for this example is shown in Fig. 20a.

Causal assumptions: $Y$ represents a favourable outcome in a drug de-addiction program within 6 months. $X$ indicates a decision made by an experienced program officer about whether to send the program participant for intensive counseling sessions with a specialized therapist.

Decisions can be made under three data-collection modes, with data values as follows:

a. Observational (Fig. 20a): the program officer follows their intuitive judgment based on years of experience, which may be affected by unobserved factors and biases. $\mathcal{L}_1$ data reveals that $P(X=1) = 0.85$, $P(Y=1|X=0) = 0.35$, and $P(Y=1|X=1) = 0.15$.

b. Interventional (Fig. 20b): the program officer overrides their natural inclination and assigns a decision to a participant, such as using a randomizing device as in a clinical trial. $\mathcal{L}_2$ data reveals $P(Y=1; do(X=0)) = 0.605$, and $P(Y=1; do(X=1)) = 0.225$.

c. Counterfactual (Fig. 20c): the program officer first registers what they normally *would have* chosen for this participant $(X=x')$ before subjecting the unit to a fixed treatment $do(X=x)$ conditioned on $x'$. $\mathcal{L}_{2.5}$ data reveals $P(Y_{X=1} = 1|X=0) = 0.65$, and $P(Y_{X=0}=1|X=1) = 0.65$.

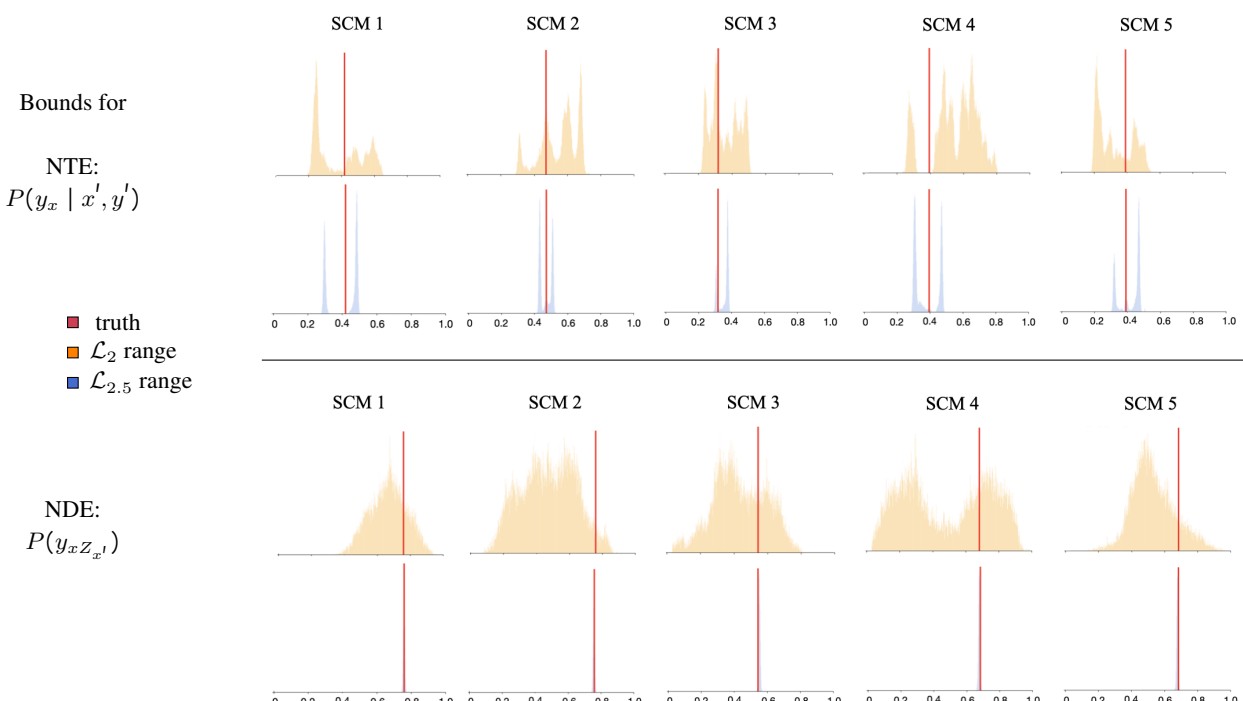

*Figure 19.* Example 2 results (over 5 random underlying SCMs) showing partial identification bounds for NTE and NDE quantities. Bounds are tighter using counterfactual data (blue) than interventional data (orange). Since NDE is identifiable from counterfactual data, blue bounds are not visible as they collapse to the true value (red).

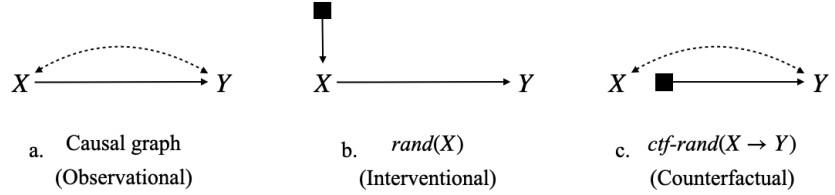

*Figure 20.* Causal diagram and different data-collection regimes for Example 3 (Unit Selection).

Following the Interventional Strategy (1), recommended by Li & Pearl (2022):

Using observational and interventional data, run the MCMC methodology described earlier to estimate the bounds for the proportion of each canonical type, $P(Y_{X=0}, Y_{X=1})$. Combine these bounds with the benefit function shown in Fig. 8 to derive the estimated bound of the avg. treatment benefit for the whole population.

Simulation using synthetic data ($N = 10^4$ samples) shows a 95% credible interval of the avg. population benefit $\Delta(1) \in [-1.3, 1.6]$.

It is inconclusive whether to administer the treatment $X = 1$ to the whole population, because it could result in net negative or positive benefit on average.

Following a Counterfactual Strategy (2):

Using counterfactual data, run the MCMC methodology described earlier to estimate the bounds for $P(Y_{X=0}, Y_{X=1}|X = x')$ - the proportion of each canonical type in the sub-population for which the program officer feels naturally inclined to assign treatment $X = x'$. Combine these bounds with the benefit function shown in Fig. 8 to derive the *conditional* subpopulation-level treatment benefit bounds.

Simulation using synthetic data ($N = 10^4$ samples) shows a 95% credible interval of the conditional benefits $\Delta(1|X = 0) \in [5.7, 11.6]$ and $\Delta(1|X = 1) \in [-2.5, -0.1]$.

The clear strategy for the program officer is to **go against their intuition** $X = x'$:

- Assign treatment $do(X = 1)$ to participants to whom they would have intuitively been inclined to reject for counseling (natural $X = 0$), since $\Delta(1|X = 0) > 0$;

- Withhold treatment $do(X = 0)$ for participants to whom they would have intuitively been inclined to recommend for counseling (natural $X = 1$), since $\Delta(1|X = 1) < 0$;

This provably dominates Strategy (1) because

$$\Delta(1) = P(X = 0)\Delta(1|X = 0) + P(X = 1)\Delta(1|X = 1) \qquad \text{Strategy 1 benefit} \qquad (26)$$
$$< P(X = 0)\Delta(1|X = 0) \qquad \text{Strategy 2 benefit} \qquad (27)$$

If the program officer chooses 0 for the whole population, they incur 0 benefit. If they choose 1 for the whole population, this would be strictly suboptimal than choosing 1 only for the subpopulation with natural $X = 0$ (Eq. 27).

### D.3. Real-world datasets

We replicate the findings in Example 2 using two public datasets, both satisfying the *instrumental variable* graph shown in Fig. 21a. (1) In the *Oregon Health Insurance* dataset (Finkelstein et al., 2011) where $Z$ denotes lottery selection, $X$ Medicaid enrollment, $Y$ mental health improvement, we want to estimate bounds for the counterfactual $P(Y_{X=1} = 1 \mid Z = 0, Y = 0)$; and (2) In the *Project STAR* dataset (Krueger & Whitmore, 2000) where $Z$ denotes assignment to a small class, $X$ enrollment, $Y$ academic performance, we want to estimate bounds for the counterfactual $P(Y_{X=1} = 1 \mid Z = 0, X = 0)$.

We fit a causal model to the data, and compute 95% *ci* of identification bounds (dotted lines) using data from $\mathcal{L}_2$ and $\mathcal{L}_{2.5}$. We get $\mathcal{L}_{2.5}$ data by simulating a *ctf-rand*$(X \to Y)$ intervention. For both cases, bounds using counterfactual data (blue) are significantly narrower than using $\mathcal{L}_2$ data (orange). For Project STAR, the blue bounds collapse to the true value (in red). This validates our identification results and Alg. 2, since the query can indeed be point-identified using counterfactual data.

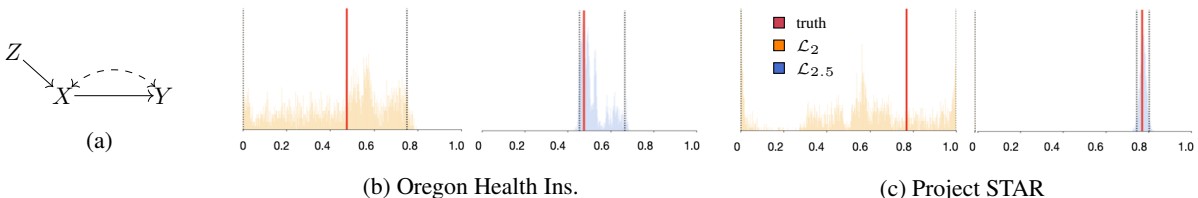

(a)   (b) Oregon Health Ins.   (c) Project STAR

*Figure 21.* (a) Causal graph for Example 3; (b), (c) estimated identification bounds for counterfactual query using $\mathcal{L}_2$ data (orange) and $\mathcal{L}_{2.5}$ data (blue) from two public datasets. Query in (c) becomes identifiable using *ctf-rand()* causing blue bounds to collapse to true value (in red).

## E. Proofs of Results

For proofs of all the results, refer to the full technical report (Raghavan & Bareinboim, 2026).

## F. Indexing an Input Data Distribution

This section is not strictly needed to understand our main results. Thm. 3.5 works with any way of writing each input data distribution as an un-nested $\mathcal{L}_3$ expression. Here, we provide a systematic way to translate from an intuitive index for each data distribution (using the actions taken in the data-collection regime) into an un-nested $\mathcal{L}_3$ expression. Any other equivalent expression would also work.

We index an input data distribution by the physical actions $\mathcal{A}$ that the experimenter takes in order to collect data. For instance, Fig. 22(Left) illustrates the observational regime, corresponding to $\mathcal{A} = \varnothing$. Fig. 22(Center) illustrates an interventional regime, where the experimenter performs a standard randomized intervention on $X$, $\mathcal{A} = \{rand(X)\}$. Fig. 22(Right) illustrates a counterfactual data-collection regime, where the experimenter performs a counterfactual randomized intervention on $X$, $\mathcal{A} = \{ctf\text{-}rand(X \to Y)\}$. See Sec. 2 for the definitions of these actions.

---

**Algorithm 4** REGIME-REGEX

---

1: **Input:** Causal diagram $\mathcal{G}$; actions $\mathcal{A}$ which index the input data distribution

2: **Output:** Un-nested $\mathcal{L}_3$ expression $P^{\mathcal{A}}(\mathbf{V}_\star = \mathbf{v})$ for the distribution of samples drawn under action set $\mathcal{A}$

3: Initialize empty conjunction $\mathbf{V}_\star = \varnothing$

4: **for** each $V \in \mathbf{V}$ **do**

5:     Initialize a potential response $V_{[.]}$, with empty subscript

6:     **for** each intervention $a \in A$ **do**

7:         $X \leftarrow$ variable intervened upon in $a$

8:         $x_a \leftarrow$ fixed value assigned to $X$ under $a$

9:         $\mathbf{C} \leftarrow$ (subset of $Ch(X)$ affected by $a$) $\cap An(V)$

10:         $\mathbf{C}' \leftarrow (Ch(X) \setminus \mathbf{C}) \cap An(V)$

11:         **for** each $C \in \mathbf{C}$ **do**

12:             **if** $a$ is superseded by a previous $a'$ involving $(X, C)$ **then**

13:                 Skip $C$

14:             **end if**

15:             **if** $C = V$ **then**

16:                 Add or replace $x_a$ in the subscript of $V_{[.]}$

17:             **else if** $C \neq V$ **then**

18:                 Add or replace $C_{x_a}$ in the subscript of $V_{[.]}$

19:             **end if**

20:         **end for**

21:         **for** each $C' \in \mathbf{C}'$ **do**

22:             **if** encountered a previous $a'$ involving $(X, C')$ **then**

23:                 Skip $C'$

24:             **end if**

25:             **if** $C' \neq V$ **then**

26:                 Add or replace $C'$ in the subscript of $V_{[.]}$

27:             **end if**

28:         **end for**

29:     **end for**

30:     Add clause $V_{[.]} = v$ to conjunction $\mathbf{V}_\star = \mathbf{v}$

31: **end for**

32: Apply consistency property to $P(\mathbf{V}_\star = \mathbf{v})$ to get un-nested $P(\mathbf{V}'_\star = \mathbf{v})$

33: Return $P(\mathbf{V}'_\star = \mathbf{v})$

---

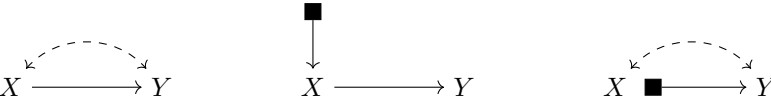

*Figure 22.* Data-collection regimes corresponding to (Left) $\mathcal{A} = \varnothing$; (Center) $\mathcal{A} = \{rand(X)\}$; (Right) $\mathcal{A} = \{ctf\text{-}rand(X \to Y)\}$.

Note that the regime in Figure 22(center) corresponds precisely to the sub-model $\mathcal{M}_x$, where a $do(x)$ intervention replaces the equation $f_X$ with a constant value $x$. However, the regime in Figure 22(right) cannot be defined in terms of a sub-model. Next, we provide a subroutine (Alg. 4) for systematically mapping the distribution index $\mathcal{A}$ to a **un-nested counterfactual regular expression**, corresponding to the distribution $P^{\mathcal{A}}(\mathbf{v}_\star)$, i.e., the distribution of variables sampled under this regime.

*Example.* Consider the graph $\mathcal{G}$ in Figure 23, being subjected to a regime indexed by actions $\mathcal{A} = \{ctf\text{-}rand(X \to Y), ctf\text{-}rand(X \to W)\}$, as illustrated. The intermediate output of REGIME-REGEX$(\mathcal{G}, \mathcal{A})$ at Line 31 would be $P(X, T, W_{x'}, Z_{W_{x'}}, Y_{xTW_{x'}})$.

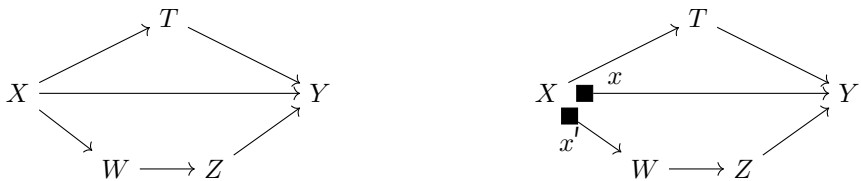

*Figure 23.* Example for regular expression under a regime (right) involving actions $\mathcal{A} = \{\textit{ctf-rand}(X \to Y), \textit{ctf-rand}(X \to W)\}$.

The final output of REGIME-REGEX$(\mathcal{G}, \mathcal{A})$ would be the expression

$$P(X = x'', T = t, W_{x'} = w, Z_w = z, Y_{xtw} = y) \tag{28}$$

■

**Proposition F.1.** *Given an input data distribution indexed by a set of physical actions $\mathcal{A}$, Alg. 4 produces an un-nested Layer 3 expression $P(\mathbf{V}_\star = \mathbf{v})$, corresponding to this data distribution.*

*Proof.* Note that Alg. 4 involves at most one level of nesting in the counterfactual expression $P(\mathbf{V}_\star = \mathbf{v})$ after Line 31. The consistency property (Correa & Bareinboim, 2025, Lemma 2.1) shows that, for any $X, Y$,

$$X_\star(\mathbf{u}) = x \implies Y_{\ldots[X_\star]}(\mathbf{u}) = Y_{\ldots[x]}(\mathbf{u}) \tag{29}$$

A straightforward application of the consistency property to $P(\mathbf{V}_\star = \mathbf{v})$ yields the equivalent un-nested $P(\mathbf{V}_{\star'} = \mathbf{v})$. ■

**Note on indexing values**: in each probability distribution expression, in general (unless otherwise stated), value terms in the main line and in the subscript are indices which can overlap. For instance, $P(x', y_x)$ refers to the distribution $P(X, Y_x)$. The specific quantity $P(x, y_x)$, where both the $x$ values are the same, can be obtained directly from one of the lines of this distribution table. We omit this level of granularity throughout the paper for readability.

## G. Frequently Asked Questions

Q1. **Where is the causal diagram coming from? Is it reasonable to expect the data scientist to create one?**

**Answer**. First, the assumption of the causal diagram is made out of necessity. The causal diagram is a well-known flexible data structure that is used throughout the literature to encode a qualitative description of the generating model, which is often much easier to obtain than the actual mechanisms of the underlying SCM (Pearl, 2000; Spirtes et al., 2000). The goal of this paper is not to decide which set of assumptions is the best but rather to provide tools to perform the inferences once the assumptions have already been made, as well as understanding the trade-off between assumptions and the guarantees provided by the method.

Second, the true underlying causal diagrams cannot be learned only from the observational distribution in general. There almost surely exist situations that two SCMs induce the same observational distribution but are compatible with different causal diagrams (see Bareinboim et al. (2022, Sec. 1.3) for details). With higher layer distributions (such as distributions from $\mathcal{L}_2$), it is possible to recover a more informative equivalence class of diagrams that encode additional constraints present in the input layer (Li et al., 2023; von Kügelgen et al., 2023).

Q2. **What is the complexity of the CTFIDU$^+$ algorithm?**

**Answer**. CTFIDU$^+$ runs in $O(zn^2(n+m))$ time, where $n, m, z$, and $d$ refer to the number of nodes, edges, (different) interventions in $\mathbf{Y}_\star$, and maximum cardinality of any observable variable in $\mathcal{G}$, respectively. See App. B.3.

Q3. **What is novel about this algorithm? Can one not use inference rules like the *counterfactual calculus* or *do-calculus* to identify counterfactuals?**

**Answer**. The scope of CTFIDU$^+$ allows for a data scientist to additionally provide as input physically realizable $\mathcal{L}_3$ data. This allows more quantities to be identified. It also subsumes previous algorithms which assume access to only $\mathcal{L}_2$ data, since observational and interventional data belong in the scope of input, too.

Indeed, the recent development of the counterfactual (ctf-) calculus (Correa & Bareinboim, 2025) provides a powerful set of inference rules to infer counterfactual queries from counterfactual (or any other) input distributions. However,

what's missing is a complete method for applying these rules in a systematic way. In fact, since CTFIDU$^+$ makes use of the ctf-calculus in its steps, Thm. 3.5 provides proof that ctf-calculus is indeed complete for the task of identifying $\mathcal{L}_3$ quantities from physically realizable data. Prior results have only shown completeness for a scope of $\mathcal{L}_2$ input data.

Q4. **What is meant by *realizable* data distribution? Is it realistic to assume access to counterfactual data?**

**Answer**. *Realizable* data distributions are those from which an experimenter can collect data samples directly using the following actions: passive observation of a system, standard interventional randomization of some variable(s) which we denote by *rand()*, or counterfactual randomization of some variable(s) which we notate as *ctf-rand()*. See Sec. 2 for definitions of these actions.

Conventional wisdom has long assumed that data can only be gathered in the real world (i.e. not in a simulated environment where the full SCM specification is known) from observational or interventional distributions. An emerging thread of research has challenged this belief, showing there are indeed realistic settings that permit *counterfactual* data collection (Bareinboim et al., 2015; Zhang & Bareinboim, 2022; Forney et al., 2017; Yang & Bareinboim, 2025) via the procedure of *ctf-rand()*.

We argue this is increasingly relevant in generative AI pipelines or automated business workflows, which allow more controllability and granularity of interventions. As prior work points out, one of the aims of such results is precisely to stimulate novel experiment design ideas in this new area of causal inference (Raghavan & Bareinboim, 2025). For instance, the increasingly automated HR pipeline in companies suggests opportunities for targeted interventions to randomize demographic details in virtual interviews, in standardized aptitude tests, or in performance-evaluation systems for remote workers, to track fairness metrics.

