# OpenReview forum: "Causal Identification from Counterfactual Data: Completeness and Bounding Results"
_ICML.cc/2026/Conference — ICML 2026 regular_

### Official Review · Reviewer_XZMx · 2026-03-11

**Soundness:** 3
**Presentation:** 2
**Significance:** 2
**Originality:** 3
**Overall Recommendation:** 5
**Confidence:** 2

**Summary:**

This paper extends the idea of counterfactual realizability and characterizes the identifiable query with access to some layer 3 data. The authors developed the IDENTIFY+ and CTFIDU+ algorithms for identifying causal and counterfactual queries with counterfactual data. The CTFIDU+ algorithm is shown to be complete. The authors also show that the fundamental limit of identifiability of layer 3 queries. For non-identifiable queries, such as NTE, the authors developed the partial identification results given data from different layers.

**Compliance With Llm Reviewing Policy:**

Affirmed.

**Final Justification:**

The authors explained the significance of this work in the rebuttal and resolved my main concerns.

**Key Questions For Authors:**

1. I'm a bit confused with the notation of $x\setminus Z$ and $x\cap Z$, could you please explain this more?
2.  For the nested counterfactual query $Y_{xZ_x'}$, $Z$ is fixed to the value it would have taken had $X$ been fixed as $x'$. Does this mean the value of $Z$ when intervened on $x'$? Is $Z$ still random after fixing $x'$?
3. In proposition 4.1, does input data regimes mean data from different PCH?

**Limitations:**

The paper did not explicitly discuss the limitations.

**Strengths And Weaknesses:**

**Strength**
- This paper proposed algorithms that generalize previous id algorithms with counterfactual queries and prove their completeness.
- The fundamental limit of the counterfactual query is quite an interesting result.
- There are many nice illustrations in the paper that help understand the key ideas.

**Weakness**
- The paper could be more self-contained. There is some terminology and notation in the paper that are a bit hard to understand without prior knowledge. For example, the nested counterfactual query and the query from Layer 2.25.
- It is not very clear the significance of the paper's contribution since the refined PCH was discussed in the existing works.


Some typos

l85 "indentification"
l89 "from in an experiment setting"

---

> ### Author Rebuttal · Authors · 2026-03-31
>
> We thank the reviewer for their observations and positive feedback. Noted on the typos.
>
> **The main significance of our work** is threefold:
>
> (A) Thm. 2.1 gives, to our knowledge, the **first completeness result in causal inference with counterfactual data**: any sound solution to this problem is equivalent to the output of Alg. 2. Counterfactual experiments are a new paradigm with many open theoretical questions and promising applications, as other reviewers also note.
>
> (B) Our new machinery (Def. E.8; Thms. E.9, E.12) provides a **unifying theoretical framework** for prior work, as noted by PP2Y.
>
> (C) Prior refinements of the PCH concerned counterfactual *realizability* (i.e. data-collection capabilities). Those prior works explicitly framed the open the question of what relationship there is, if any, between realizability and counterfactual *identification* (i.e. computing counterfactuals). Sec. 3 answers this by proving a **fundamental duality result (Thm. 3.1, Cor. 3.2): nonparametric identification of causal quantities is essentially trying to mimic realizability**. If a counterfactual query can never be physically realized, then the graph contains some confounding that cannot be removed even by sophisticated identification methods. This can prevent wasted effort searching for impossible experiment designs.
>
> This framework also suggests new research directions: e.g., when identification is blocked by a specific term in the “causal lattice” (App. C.3), one can impose assumptions only on that bottleneck term (e.g., monotonicity, linearity, Lipschitz sensitivity). Our paper provides the needed foundation to do this.
>
> ----
>
> We additionally clarify some concepts below, as requested.
>
> (1) **Notation**: Suppose random variables $\mathbf{X}$ = {$T, Z$} and value vector $\mathbf{x}$ = {$t, z$} , then $\mathbf{x} \setminus Z$ = {$t$} and $\mathbf{x} \cap Z$ = {$z$}. That is, we take the subset of values corresponding to overlap with $Z$ or its complement.
>
> (2) **Nested counterfactuals**: For a fixed unit $U=u$, the potential outcome $Z_{x’}(u)=z$ under do($x’$) is a constant. The nested potential outcome $Y_{x Z_{x’}}(u)$ refers to the counterfactual world where we perform do($X=x$) and do($Z=Z_{x’}=z$) for this unit $u$. You are right that $Z_{x’}$ is a random variable, so the value of $Y_{x Z_{x’}}$ is set equal to $Y_{xz}$, $Y_{xz’}$, $Y_{xz’’}$, etc., depending on the hidden noise $u$.
>
> This formalism was introduced to study path-specific effects; see [1] for a classic treatment using a slightly different notation. Suppose there are two paths from $X$ to $Y$ as in Fig 1a, we may ask how the total effect of $X$ on $Y$ decomposes into *direct* and *indirect* effects (via a mediator $Z$). If the direct effect, defined as $P(y_{x Z_{x’}})-P(y_{x’})$, is 0, then the avg. treatment effect $P(y_x)-P(y_{x’})$ is fully explained by the indirect pathway through the mediator.
>
> (3) **Prop 4.1**: The input data collections $\mathbb{A}, \mathbb{A}’$ are both collections of data distributions from the *same* PCH. I.e., they both contain distributions from the same underlying SCM under different interventions. E.g. $\mathbb{A}$ could contain only observational distributions while $\mathbb{A}’$ could contain observational + some interventional distributions, taken from the *same environment*.
>
> **Layer 2.25**:
>
> This refers to queries whose distributions can be physically realized using ctf-rand(), **but** with the restriction that for each variable $X$ we may perform at most one randomization, and that randomization must affect all children of $X$, as in Fig. 4b. We are not allowed to perform *path-specific randomizations* like Fig. 4c or Fig. 1a. We provide more background in Appendix C.2 and lines 1121–1131. For this paper, L2.5 is the critical concept; L2.25 is less central to the results.
>
> ---
>
> We are happy to clarify any further details of our results or the background concepts. In light of our contributions, if we have addressed the main concerns, we respectfully ask the reviewer to consider improving their score or confidence.
>
> ---
>
> **References**:
>
> [1] Pearl (2000), *Causality*, Sec. 4.5.2

---

> > ### Author Rebuttal · Reviewer_XZMx · 2026-04-02
> >
> > Thanks the authors for the clarification. My concerns are adequately resolved. I would like to update my score.

---

> > > ### Author Response · Authors · 2026-04-07
> > >
> > > We thank the reviewer for the helpful suggestions, and appreciate the productive discussion!

---

### Official Review · Reviewer_mrCe · 2026-03-11

**Soundness:** 3
**Presentation:** 4
**Significance:** 2
**Originality:** 3
**Overall Recommendation:** 4
**Confidence:** 4

**Summary:**

This article makes contributions to Layer 3 (PCH) causal identification. Through counterfactual realizability, the authors advanced ctfID (Correa et al. (2021)), assuming access to a subset of Layer 2 data to their ctfIDU+ algorithm, assuming known a subset of realizable Layer 3 data (Table 1). This new algorithm guarantees counterfactual identification under scenarios implementable experiments with given counterfactual distributions and new data. The authors proved the completeness of their algorithm. The paper demonstrates the significant performance of search-based identification of tightened boundaries through counterfactual data, even when causal identification is difficult to achieve, using simulations and synthetic data.

**Compliance With Llm Reviewing Policy:**

Affirmed.

**Key Questions For Authors:**

The framework assumes practitioners can correctly classify whether their available data belongs to which layer. In practice, how should a researcher determine, for example, whether they have truly collected the specified layer data with some confounding? What diagnostic tests or validation procedures would you recommend to verify which layer one's data actually comes from, especially given that the underlying SCM is unknown? Your completeness result (Theorem 2.1) assumes the causal diagram G is correctly specified. How sensitive are your identification results to misspecifications of the graph, particularly to incorrectly omitted or included bidirected edges?

**Limitations:**

The paper provides theoretical definitions of layers but no practical guidance for determining which layer a dataset belongs to. The causal diagram is assumed known and correct, which is rarely true in practice (note this applies to many PCH settings, not particularly for this submission). No real-world case studies demonstrating the methods work outside carefully controlled simulations.

**Strengths And Weaknesses:**

Soundness. Theorem 2.1 (completeness). It is a significant advance in counterfactual identification. The algorithm goes beyond those algorithms dealt with Layer 2 data, moving forward to Layer 3.

Presentation. The paper has a clear exposition and rigorous logic. The motivations, problem definitions, algorithm illustrations, theoretical justifications and experiments are well connected. Although with complex theoretical proofs, the author efficiently explained their core concepts through figures, tables and examples.

Significance. Developed mathematical proofs for their algorithms (Theorems 2.1, 3.1, etc.) Illustrated the algorithm with synthetic examples to demonstrate how they would work. Showed that their bounds are provably tighter (Proposition 4.4) through mathematical derivation.

Originality. Theorem 3.1 and Corollary 3.2 illustrate the identifiability and realizability of counterfactual query and their relationship, point out the limit of identification. These results provide guidance of future research directions and some theoretical bases, correct potential misunderstandings of Layer 3 identification with advanced methods.

Weaknesses: The paper provides theoretical definitions of layers but no practical guidance for determining which layer a dataset belongs to. The causal diagram is assumed known and correct, which is rarely true in practice (note this applies to many PCH settings, not particularly for this submission). No real-world case studies demonstrating the methods work outside carefully controlled simulations.

---

> ### Author Rebuttal · Authors · 2026-03-31
>
> We thank the reviewer for their positive feedback and the helpful suggestions. We appreciate the recognition of the novelty of our results in the literature, the significance for future causality research directions, and the clarity of presentation of our proof concepts.
>
> We address the three main concerns:
>
> **1. Unclear how to determine the PCH layer of a query**
>
> This is a valid point. We exactly answer this question in Thm. C.3: a query belongs to Layer 2.5 iff the counterfactual ancestor set (Def. B.2) of the Potential Outcomes (PO) in the query does not contain multiple POs of the same variable. Otherwise, it belongs to Layer 3 \ 2.5.
>
> In Fig 3, consider the queries $P(y_x \mid z_{x’}, x’’)$ and $P(y_a \mid z_{a’}, a’’)$. The counterfactual ancestor set of {$Y_x, Z_{x’}, X$} is {$Y_x, Z_{x’}, X, A_x, A_{x’}$} which contains both $A_x, A_{x’}$, so the first query is outside Layer 2.5. By contrast, the counterfactual ancestor set of {$Y_a, Z_{a’}, A$} is {$Y_a, Z_{a’}, A, X$}, with no repeated POs, so the second query lies in Layer 2.5.
>
> There is a similarly simple condition for checking if an expression lies in Layer 2.25 (Def. 16 and Lem. 2 in [1]). We will move both of these to our main body.  Of course, if all POs share the same subscripts, the expression is in Layer 2; if no PO has subscripts, it is in Layer 1.
>
> As PP2Y notes, an important takeaway is that if a complex counterfactual query does not lie in L2.5, then nonparametric point identification is impossible. This can prevent wasted effort searching for an experiment design to achieve identification.
>
>
> **2. Assuming knowledge of the causal graph**
>
> As the reviewer observes, this is a known limitation in the field. Causal graphs are a standard assumption used to make progress in a lot of causal inference tasks.
>
> Ideally, the pipeline would begin with graph discovery methods that encode edge uncertainty, e.g. using PAGs representing an equivalence class of graphs compatible with the data. Another common practical approach is to place a Bayesian prior over possible graphs and computed weighted average identification values across them; the spread of values then reveals sensitivity to graph specification. We did not discuss this because we felt it would be viewed as peripheral.
>
> Our main point is that *even if we commit to a graph*, there was previously *no general method for identifying target queries from counterfactual data*. This is the gap our work fills.
>
> **3. Experiments on real-world data**
>
> We stress that our simulations are fully rigorous. We did not cherry-pick parameters; we repeated the methodology over random underlying SCMs (Fig. 17) and provided code for reproducibility. Nevertheless, we also applied the same methods to public datasets as suggested:
>
> - **Oregon Health Insurance** [2]: we compute empirical identification bounds for a counterfactual query and show that bounds are tighter with L2.5 data than with L2/L1. We simulate ctf-rand( ) interventions to obtain L2.5 data [(url)](https://i.ibb.co/CsTv49nN/Figure1.png).
> - **Project STAR** [3]: we evaluate a query which is identifiable using ctf-rand(); bounds collapse conditioned on L2.5 data, thus validating Alg. 2, but not given L2/L1 data [(url)](https://i.ibb.co/C3S23Gh6/Figure2.png)
>
> We hope this reassures the reviewer that the findings are robust across different data-generating environments.
>
>
> **Main contributions**
>
> We reiterate the significance of our work.
>
> (A) Thm. 2.1 gives, to our knowledge, the first completeness result in causal inference with counterfactual data: any sound solution to this problem is equivalent to the output of Alg. 2. Counterfactual experiments are a new paradigm with many open theoretical questions and promising applications, as other reviewers also note.
>
> (B) Our new machinery (Def. E.8; Thms. E.9, E.12) provides a unifying theoretical framework for prior work, as noted by PP2Y.
>
> (C) Sec. 3 proves a fundamental duality result (Thm. 3.1, Cor. 3.2): nonparametric identification of causal quantities is, in essence, mimicking realizability/data collection.
>
> This framework also suggests new research directions: e.g., when identification is blocked by a specific term in the “causal lattice” (App. C.3), one can impose assumptions only on that bottleneck term (e.g., monotonicity, linearity, Lipschitz sensitivity). Our paper provides the needed foundation to do this.
>
> In light of these points, we respectfully ask the reviewer to consider improving their score if we have addressed their concerns. We are happy to address any follow-up questions!
>
> **References**
>
> [1] Yang & Bareinboim (2025). *A hierarchy of graphical models for counterfactual inferences*
>
> [2] https://www.nber.org/research/data/oregon-health-insurance-experiment-data
>
> [3] https://rdrr.io/cran/AER/man/STAR.html

---

> > ### Author Rebuttal · Reviewer_mrCe · 2026-04-03
> >
> > Thanks for addressing my questions. From the rebuttal, the mentioned real data analyses are reasonable, but still not with concrete identifications of causal completeness.

---

> > > ### Author Response · Authors · 2026-04-07
> > >
> > > Thanks to the reviewer for the comments and for the insightful discussion! And for acknowledging that the other questions have been addressed, particularly the main objection raised about **how to assign PCH layer membership**.
> > >
> > > Regarding the experiments, there were a few different critiques in the review:
> > >
> > > 1. "*The original experiments were run on simulated data and not real-world data*": we infer from the acknowledgment that this has **now been addressed**, as we re-ran our experiments on real-world data.
> > >
> > > 2. "*The new experiments do not address concrete identification*": actually, **we do address point-identification using the Project STAR** dataset - pls refer to the figure linked in our rebuttal. This query is not identifiable from L1 and L2 data (thus the bounds are wide in yellow and orange), but it becomes identifiable when we have data under ctf-rand($X \to Y$), causing the L2.5 bounds (blue plot clearly not visible) to collapse to the true value shown in red; thus **validating Algorithm 2**, because the algorithm would return an ID expression here.
> > >
> > > 3. "*The new experiments do not address algorithmic completeness*": to be fair, **this is a new critique about completeness that is being raised now** and we emphasize that we cannot answer any follow-up questions on this. We originally didn’t show plots for completeness because (as *PP2Y* also notes) showing point-identifiability on an example wouldn’t be an interesting use of limited space.
> > >
> > > As the reviewer requests, we demonstrate the following sanity check for algorithmic completeness! **Note**: by design, we need to use random simulated SCMs because the main point of “completeness” is that *for all possible underlying causal models compatible with the given causal graph, the identification formula gives the correct answer; furthermore, where Alg. 2 FAILS, we can guarantee non-identifiability*. This shows that our method works for any valid ground-truth SCM.
> > >
> > > The pipeline is quite straightforward:
> > >
> > > - We generate 5 random causal graphs having 3-10 variables, with arbitrary edges, arbitrary confounding structure.
> > >
> > > - For each graph, we manually specify (a) some *input L2.5 data*; (b) a *query that is identifiable* from this data, with the identification formula given by Alg. 2; (c) a *query that is not identifiable* from this data.
> > >
> > > - For the identifiable query in (b), we generate *5 random causal models* and show that in each model, the **identification formula given by Alg. 2 (blue column in figures) exactly matches the true value (green column in figures)**; these can be computed exactly from the SCM.
> > >
> > > - For the non-identifiable query in (c), through rejection-sampling we generate *2 causal models* that **agree on input and disagree on output distributions (a standard way of proving that a query is non-identifiable)**.
> > >
> > > - Thus, we show that our identification method is **sound and complete, using 25 + 10 SCMs in total** (these numbers can be scaled arbitrarily - this was chosen merely for ease of depiction).
> > >
> > > Links to figures using random graphs numbered 1-5: [[1](https://i.ibb.co/dwMYg057/Fig1.png)], [[2](https://i.ibb.co/SXXcW6z5/Fig2.png)], [[3](https://i.ibb.co/6cjDcyTz/Fig3.png)], [[4](https://i.ibb.co/2160gfWz/Fig4.png)], [[5](https://i.ibb.co/k6Js8Tgk/Fig5.png)]. We will add these to the appendix - thank you for the suggestion!
> > >
> > > We request the reviewer to kindly check whether all the critiques have now been addressed. Thank you!

---

### Official Review · Reviewer_PP2Y · 2026-03-13

**Soundness:** 3
**Presentation:** 4
**Significance:** 3
**Originality:** 2
**Overall Recommendation:** 5
**Confidence:** 3

**Summary:**

The paper studies the problem of counterfactual identification in the counterfactually realizable setting - i.e., where counterfactual data can be experimentally collected. It builds on recent work which introduced counterfactual randomization experiments (through the notion of operations like "ctf-rand"), and addresses which quantities are identifiable in such settings. In particular, the authors develop identification algorithms (IDENTIFY+, CTFIDU+) which essentially extend the classical analogues (IDENTIFY, CTFID) to this setting, and prove completeness. They then prove that not all quantities are identifiable in this settings, establishing that the "frontier" of point identification corresponds to $L_{2.5}$. This motivates them to show that $L_{2.5}$ data can still be used to tighten partial identification bounds for quantities in $L_3$ but not $L_{2.5}$, and they provide such tightened bounds for the NTE in a bow graph. They evaluate their bounds on two case studies (traffic camera + unit selection).

**Compliance With Llm Reviewing Policy:**

Affirmed.

**Final Justification:**

I appreciate the authors clarifying my concerns about the proof techniques and handling of continuous variables in the rebuttal.

My final recommendation and opinion of the paper has not changed since the rebuttal - the paper extends complete identification algorithms to a relatively new and interesting "counterfactual realizability" setting. While this setting is somewhat rare, it is a valuable contribution to researchers who work with data in this setting and I don't see a reason it should not be accepted.

**Key Questions For Authors:**

1. In example 1, the authors state that :

*"X might affect Y due to a high correlation in training data between car color preferences and a driver’s socioeconomic group (a good predictor for speeding)"*.

This sounds more like a confounder between $X$ and $Z$, rather than the direct edge $X \to Y$. The direct edge $X \to Y$ can surely only represent artefacts of how certain car colors may affect camera triggering (since this effect holds speeding fixed)?

2. Could one strengthen Prop 4.1 to give a strict inclusion on the interval? what sorts of conditions would be needed on $A'$ and $A$?

3. I would appreciate it if you could clarify the proof steps in Thm E.9 mentioned above.

4. Do continuous variables represent a fundamental obstruction to the identifiability techniques here or can one formulate measure-theoretic analogues? (e.g., by replacing $P(T^* = t)$ with $P(T^* \in \mathcal A)$  etc.)

**Limitations:**

**Discrete Variables**: The current framework only handles discrete variables. It is not clear to me whether continuous variables present a fundamental obstruction to the proof machinery or whether it is just a matter of convenience to work with sums algorithmically. Either way, at present things can only be applied out the box for the jointly-discrete case, which is a notable limitation.

**Strengths And Weaknesses:**

**Soundness**

- **Mathematical Framework:** The theoretical approach seems to build on a well-established framework for identification results. The mathematical arguments seem correct and appear to use standard constructions/approaches as in prior identification work (e.g., completeness via graphical constructions such as hedges).
- **Algorithms and Definitions**: All mathematical definitions and constructions are clearly specified (both in the main text and the appendix). I did not notice any logical inconsistencies in the main algorithms (IDENTIFY+, CTFIDU+).
- **Evaluation**: The paper only evaluates the partial identification bounds on the NTE derived in Sec 4. However, I think this is fine because the other main results are all point-identifiability results or completeness results, and so implementing them on an example would likely not strengthen the paper much further. Since the Prop 4.1 and 4.4 leave open the question of how much NTE bounds can be tightened in practice by conditioning on $L_{2.5}$ data, focusing the evaluation/examples on these bounds is the right move here.
- **Identifiability Proof Steps**: Some proof steps could be made more explicit for those less well-versed in the graphical constructions/techniques used in the identification proofs. For instance, the proof of the completeness Thm 2.1 requires identifiability of $P(C_*=c)$ from $P(T_* = t)$ to be equivalent to IDENTIFY+ returning an expression for it (Thm E.10). A crux of the proof of this equivalence boils down to Thm E.9, which shows ctf-hedges are not identifiable.  However, I could do with clarification on a couple of steps in the proof of Thm E.9:

1. The "value chaining" argument for the consistency property to get $P(T_* = t) = P(T=t)$ could do with a bit more explanation. In particular, to chain together consistency assumptions, wouldn't we need to guarantee that the parent assignments in every subscript are the same as the realized parent values in ($P$-almost) all events? If so, how do we know this is satisfied here?
2. the "w.l.o.g. take $\sum c = 0$ (mod 2)" could be better explained. I assume this technique has come from Shpitser and Pearl (2006, Thm 4) (since Thm E.9 adapts it), but it wasn't clear to me why one can restrict attention to such values of $c$.

**Presentation**

- I find the main text very well-structured and clearly written. The motivation and counterfactual realizability set-up was well-explained, and the additional background provided in Appendix C was very helpful.
- Some of the notation is quite cumbersome, but I don't see obvious ways around this. The additional examples in the appendix for the provided algorithms are used well to add deeper intuition for what is going on (e.g., for IDENTIFY+ Appendix B.4 is helpful to clear up that it is not simply marginalizing out terms).
- As mentioned above, some proof steps could be explained more clearly in the Appendix, particularly as the proofs rely heavily on quite specific constructions like (counterfactual) hedges and forests.

**Significance**

I think this paper makes several meaningful contributions to the causal identification literature, by pushing forward the frontier in an interesting and relatively new setting (i.e., counterfactual realizability):

- The main contribution is developing a (complete) identification algorithm (CTFIDU+) that can be used to determine whether  counterfactual queries of interest are identifiable, and usefully returns the estimand itself when it is. This, in my view, could significantly impact practitioners in systems where the generative process can be partially controlled so-as to generate counterfactual data --- especially since the types of expressions one can construct here can be quite complicated (involving nested counterfactuals etc.), and the counterfactual realizability setting has only been recently formalized. The ability to automatically 'plug-and-play' counterfactual queries into such an algorithm to recover estimands where identifiable is particularly appealing.
- The characterization of the identification frontier is also useful in setting the stage for which types of quantities can be identified using such algorithms. In particular,  the fact that $L_3$ quantities that are not in $L_{2.5}$ are not identifiable given $L_{2.5}$ data is very useful as it prevents practitioners from blindly searching for counterfactual data that can be used to identify such quantities where it is fundamentally not possible.
-  Prop 4.1 is helpful in showing that counterfactual data may tighten bounds, but the fact that the interval inclusion is not strict limits its usefulness. The partial identification bounds for NTE are also clearly useful (as demonstrated by the applications), but remain specific to this quantity.

In my view, the wider limitations to significance are: (i) the counterfactual realizability setting remains (to my understanding) somewhat rare due to the required generative process control, and (ii) at present only jointly discrete variables are handled.

**Originality**

- The paper mostly extends concepts and machinery from previous work (i.e. counterfactual realizability and the PCH rung definitions including $L_{2.25}$, $L_{2.5}$ all come from earlier work, while the IDENTIFY+ and CTFIDU+ algorithms are essentially extensions of analogous algorithms used in the non-counterfactual-realizability cases).
- Having said that, counterfactual realizability has only been recently formulated and there is little work in this setting, so I still consider the contributions here somewhat novel.

---

> ### Author Rebuttal · Authors · 2026-03-31
>
> We thank the reviewer for the positive feedback about the significance of our results to the literature and to future research directions. We appreciate the highly engaged questions!
>
> **Applications**
>
> We believe there are exciting settings where ctf-rand() will become increasingly relevant. Examples include: (a) GenAI workflows allowing controllability and randomized digital edits at scale; (b) interventions in automated HR pipelines to randomize demographic details in virtual interviews, or in performance-evaluation dashboards for remote workers. By expanding the scope of counterfactual identification (and its limits), we hope to stimulate new application ideas.
>
> **Handling continuous variables**
>
> This is not a limitation. If (a) variables are *jointly* continuous, (b) the relevant conditional densities are well-defined, and (c) positivity holds, then in Alg 2 we can replace sums by integrals and PMFs by PDFs (with the caveat that PDF equalities hold only "almost everywhere"). The proof can be adapted by thresholding each variable into a binary version. If the discretized PMF is non-identifiable, so is the original PDF. Then apply the Thm E.9 strategy to the binary distributions.
>
> In practice, continuous variables are often just discretized into buckets.
>
> **Counterfactual hedges and thickets - significance**
>
> Thank you for appreciating this machinery! In fact, we intend to use the extra page to better explain the *significance of counterfactual hedges and thickets*. This is not merely an analog of prior work, but a new unifying framework for identification.
>
> Background: [1] defined *C-forests* and *hedges* and showed how they certify non-identification of Layer 2 queries from observational data. A related construction was then used to prove completeness of IDC* for L3 identification given *all* L2 distributions. [2] later introduced *hedgelets* and *thickets* to show L3 non-identification given *some* L2 data, but the construction was quite complicated and the proof strategy had to handle multiple edge cases (see Defs. 5, 6 and Sec. 3.1 in [2]).
>
> Our work introduces a new hedge construction based on both graph structure and *value-assignments for potential outcomes* (Def. E.8), yielding a **much simpler proof technique that unifies prior work**. This framework could unlock new insights to identification (e.g. where to introduce minimal extra assumptions to overcome identification hurdles).
>
> **Clarifying proof details**
>
> The target of IDENTIFY+ has the form $P(X_{pa_x}=x, Y_{pa_y}=y…)$, i.e., the probability of a *specific joint value-assignment*, whereas the input is a *full distribution* $P(X_{pa_x}=x, Y_{pa_y}=y, Z_{pa_z} = z…), \forall x, y, z$. Note: the indexing of values and subscripts in the input expression is sensitive.
>
> Take Fig 5a. Suppose we want to identify $P(Y_x = y)$ for a specific $x,y$, and we have the input ctf-factors $P(Y_x=y, X=x’), \forall y,x,x’$. This is not a ctf-hedge: we can marginalize $\sum_{x’} P(y_x, x’)$ to get the target. But if we have only the input ctf-factors $P(Y_x=y, X=x), \forall y,x$, then this forms a ctf-hedge because of “value chaining” — we cannot marginalize or decompose any further. Note: $\sum_x P(Y_x=y, X=x) \neq P(y_x)$.
>
> Incidentally, $P(Y_x=y, X=x), \forall y,x,$ is equivalent by consistency to the observational $P(Y=y, X=x), \forall y,x,$ so we essentially recover the old result that ATE is non-identifiable from observational data in the bow graph. **Ctf-hedges generalize this insight across the PCH.**
>
> Regarding the bit-encoding scheme: we do know the value of $\mathbf{c}$ because it is specified in the IDENTIFY+ function call. For binary variables the bit parity $\sum \mathbf{c}$ is either 0 or 1. If it is 1, then we can just add an extra bit in the rest of the proof steps. “W.l.o.g assuming 0” simply avoids spelling out the extra case.
>
> **Prop 4.1 - strict subset**
>
> That is a great question. Guaranteeing a strict subset is an open problem - we need to establish criteria s.t. the new polytope of feasible SCM parameters must strictly exclude the earlier extremal points. In the NTE case, the range in Prop 4.4 is a strict subset iff the inequality in Eq 37 is strict. We do not see an obvious general criterion; it may depend on the specific query and graph.
>
> **Example 1 edges**
>
> For the directed edge $X \to Y$, see our response to MZvb. Note that $Y$ is the *model output* (ticket decision). Sure, we can add an extra bidirected edge between $X, Z$ and amend the example as in [this figure](https://i.ibb.co/gMX468tY/Example1.png). Graph (a) is the observational regime; (b) is data from an RCT (p. 1, lines 40–41); and (c) is a *ctf-rand()* edit on the RCT data. The point is that the extra *ctf-rand()* is needed to identify NDE.
>
>
> **References**:
>
> [1] Shpitser & Pearl (2006). *Identification of Joint Interventional Distributions in Recursive Semi-Markovian Causal Models*
>
> [2] Lee et al (2020). *General Identifiability with Arbitrary Surrogate Experiments*

---

> > ### Author Rebuttal · Reviewer_PP2Y · 2026-04-01
> >
> > Thank you for clarifying the proof steps and the example. I hear your arguments re. discretizing/thresholding continuous variables, but in my view, discretizing naturally real-valued variables is rare in practice and usually reflects an algorithmic limitation. If the algorithm/downstream task is invariant to the discretization scheme, then fine, but this is usually not true.
> >
> > However, this is a minor point that I don't want to dwell on - I consider the concerns resolved, but I would request that you discuss limitations re. continuous variables in a way that doesn't purely trivialize them by discretization.

---

> > > ### Author Response · Authors · 2026-04-07
> > >
> > > Thanks to the reviewer for the incredibly helpful suggestions, and discussion points. These questions have helped us much better position our latest draft.
> > >
> > > We appreciate the reviewer pushing us to address continuous variables. We will definitely add a discussion on limitations based on the above suggestions: **theoretically**, the algorithm will handle continuous variables as-is if we just swap PMFs with PDFs; **practically** there are potential challenges, and workarounds like discretization may change the downstream data-analysis.
> > >
> > > This is appreciated!

---

### Official Review · Reviewer_MZvb · 2026-03-20

**Soundness:** 3
**Presentation:** 3
**Significance:** 3
**Originality:** 3
**Overall Recommendation:** 4
**Confidence:** 2

**Summary:**

This paper proposes a method where they utilize observational, interventional, and a specific type of counterfactual data to estimate and sample from a family of counterfactual distributions. The authors provide a complete identification algorithm for this purpose. The authors also connect identification and realizability of causal queries from Pearl’s three layers. Finally, the authors provide experimental performance in synthetic and real-world datasets.

**Compliance With Llm Reviewing Policy:**

Affirmed.

**Final Justification:**

My concerns are addressed after the rebuttal. I increase my score from 3 to 4.

**Key Questions For Authors:**

**Questions**
1. Can ctf-rand() be said as: no causal mechanisms are changed but the inputs to a mechanism are randomized?
2. What is the intuitive explanation for this query: $P(y_x \mid z_{x'}, x'')$ and how do you obtain it from the graph?
3. What would be the sample complexity with observational, interventional, and counterfactual data to estimate a target counterfactual query?

**Limitations:**

Yes

**Strengths And Weaknesses:**

**Strength**

The paper addresses a very interesting unexplored problem. The paper is written in a well-read manner. The paper introduces multiple novel theoretical contributions such as i) the connection between identifiability and realizability ii) a complete algorithm for identifying a specific family of counterfactual distributions from available observational, interventional, and counterfactual data. The proposed approach appears theoretically sound to me.


**Weakness**

1. Example 1 needs more logical consistency. Besides that, doesn’t a causal graph represent the underlying true causal model? Even though there might be a spurious correlation between car color (X) and assigning a speeding ticket (Y), shouldn’t that be represented with a bi-directed edge instead of a causal edge?
2. The definition of nested counterfactuals is not clear. Why is it called nested? How is it different from other counterfactual queries? Some more practical examples should be provided to discuss the concept.
3. No real-world experiments were shown for the proposed method. Although the paper has some important theoretical contributions, it is important evidence required to understand what the practical utility of the proposed algorithm is. I would request the authors to show performance in an experimental setup where real observational, interventional, and counterfactual data can be collected and nested queries can be estimated from that.
4. The authors should show performance on more complex graphs and how the performance changes as the graph complexity increases.

---

> ### Author Rebuttal · Authors · 2026-03-30
>
> We thank the reviewer for the feedback. We address these concerns and clarify some scope misunderstandings. **Update:** we now provide results on 3 public datasets.
>
> **Main Contributions**
>
> (A) Thm. 2.1 gives, to our knowledge, the first completeness result in causal inference with counterfactual data: *any* sound solution to this problem is equivalent to the output of Alg. 2.
>
> (B) Our new machinery (Def. E.8; Thms. E.9, E.12) provides a unifying framework, as noted by *PP2Y*.
>
> (C) Sec. 3 proves a fundamental duality result (Thm. 3.1, Cor. 3.2): nonparametric identification of causal quantities is, in essence, mimicking realizability/data collection.
>
> This framework also suggests new research directions: e.g., when identification is blocked by a specific term in the “causal lattice” (App. C.3), one can impose assumptions only on that bottleneck term (e.g., monotonicity, linearity, Lipschitz sensitivity). Our paper provides the needed foundation to do this.
>
> **RE: Example 1**
>
> Yes, we require a direct edge $X \to Y$. $X$ may be correlated with a hidden feature (e.g., socioeconomic status) associated with outcomes in the historical data; a model trained on such data may then directly use $X$ in its prediction, $Y$. $X \to Y$ means intervening on $X$ may directly change the **model output** ($Y$). For another fairness example, see Fig. 2.1 of [1] on COMPAS data; $X \to Y$ captures the possibility that the model directly uses a banned attribute (e.g., race) in prediction. We also relax the setting to allow additional bidirected edges on p. 7, lines 363–368. This does **not** affect the paper’s main contributions.
>
> **Nested counterfactuals**
>
> We explain this in Sec. 1.1; see also [2]. In Fig. 1a, let $X$ be gender, $Y$ college admission, and $Z$ applicant’s department choice. The nested counterfactual $P(y_{xZ_{x'}})$ asks: what is the admission probability for a *male applicant* if we changed his *department choice* to whatever it would have been *had he been female*? This reveals if disparity across genders is due to direct discrimination ($X \to Y$) or indirect path $X \to Z \to Y$ (e.g., women apply to more competitive departments). Identifying such probabilities is challenging; this is precisely where our results help. Note: this is different from the joint distribution $P(y_x, z_{x'})$.
>
> **Experiments and graph scaling**
>
> These are not the main focus of the paper, but we added them at the reviewer’s request.
>
> - **Oregon Health Insurance** [3]: we compute empirical identification bounds for a counterfactual query and show that bounds are tighter with L2.5 than with L2/L1. We simulate ctf-rand( ) interventions to obtain L2.5 data [(url)](https://i.ibb.co/CsTv49nN/Figure1.png).
> - **Project STAR** [4]: we evaluate a query which is identifiable using ctf-rand(); bounds collapse under L2.5, validating Alg. 2, but not under L2/L1 [(url)](https://i.ibb.co/C3S23Gh6/Figure2.png)
> - **NSW** [5]: given a fixed sample size of 200, we show that the estimation error range increases as more variables are added to the counterfactual query [(url)](https://i.ibb.co/ZRc68JLZ/Figure3.png)
>
> Graph-scaling “performance” can mean two different things. *Computationally*, we show in App. B.3 that Alg. 2 scales as $n^2(n+m)$ for a graph with n nodes and m edges. *Statistically*, for fixed sample size, estimation becomes harder when the query involves more variables, because variance increases (see NSW experiment above).
>
> **Sample complexity**
>
> Sec. 1 (see Fig. 2) distinguishes *identification methods*—which map counterfactuals to unique functions of input distributions, the scope of our work—from *statistical estimators* of those functions from finite samples. Sample complexity concerns the latter and depends on the chosen estimator. A naive MLE would use concentration bounds term-by-term on the expression returned by Alg. 2; more advanced estimators (e.g., double ML) may improve this, but estimation is outside our scope.
>
> **Questions**
>
> 1. Yes—this is a good intuition for ctf-rand().
>
> 2. Let $X$ be college applicant’s race, $Y$ admission, $Z$ scholarship award, and $A$ the applicant’s CV. The quantity $P(y_x \mid z_{x'}, x'')$ asks: what is the probability of admission if the applicant was of race $x$, given that their actual race is $x''$ and they would have received a scholarship had their race been $x'$? Such quantities are central in fairness; Example 2 of [6] studies exactly this metric.
>
> We respectfully request your reconsideration based on the significance of our results. We are happy to answer any follow-up questions!
>
> **References**
>
> [1] Plecko & Bareinboim (2024), *Causal Fairness Analysis: A Causal Toolkit for Fair Machine Learning*
>
> [2] Pearl (2000), *Causality*, Sec. 4.5.2
>
> [3] https://www.nber.org/research/data/oregon-health-insurance-experiment-data
>
> [4] https://rdrr.io/cran/AER/man/STAR.html
>
> [5] https://users.nber.org/~rdehejia/data/.nswdata2.html
>
> [6] Raghavan & Bareinboim (2025), *Counterfactual Realizability*

---

> > ### Author Rebuttal · Reviewer_MZvb · 2026-04-02
> >
> > I thank the author for their response and additional experiments. Most of my concerns are addressed. However, I have a few additional questions based on their response.
> >
> > 1. The additional experiments use simple graphs such as the bow with an instrument or a backdoor graph. Could the authors discuss how practical is the method for real-world setups? For example, how does the method scale for a larger graph suppose 10-20 variables? What challenges will it face?
> >
> > 2. For the Oregon Health Insurance experiment, how do the authors obtain data from L2.5 layer?
> >
> > 3. For the NSW experiment, shouldn't the bound uncertainty reduce since the authors are using more observed confounder instead of none?

---

> > > ### Author Response · Authors · 2026-04-08
> > >
> > > Thanks to the reviewer for the thoughtful follow-up questions and productive discussion. We appreciate your recognition that the other concerns were resolved, and also your push to address practical considerations more directly. This helps us strengthen the paper!
> > >
> > > ---
> > >
> > > > Oregon Health Insurance experiment: how do the authors obtain data from L2.5 layer?
> > >
> > > We first fit a causal model (SCM) to the real-world data. We then use this fitted SCM as a data-generating process to simulate the *ctf-rand()* intervention and obtain L2.5 data. For robustness, we repeated the fitting step with different random seeds (yielding different SCM fits) and observed similar results. Figure 17 in our paper shows this, and the supplementary material includes code for full reproducibility. Since the ground-truth SCM is unknown in real data, these repeated fits serve as a sanity check that the conclusions are not an artifact of a particular random initialization.
> > >
> > > ---
> > >
> > > > NSW experiment: shouldn't the bound uncertainty reduce since the authors are using more observed confounder instead of none?
> > >
> > > The *Oregon Insurance* and *Project STAR* experiments are about how **identification bounds shrink with more counterfactual data**, showing the value of *ctf-rand()* in real-world settings. These results concern identification bounds, not statistical error bars.
> > >
> > > The *NSW experiment* serves a different purpose: it illustrates the **practical cost of adding more variables to the target query and graph**. The target queries are $P(Y_x = 1 \mid x')$, $P(Y_x = 1 \mid x', z_1)$, $P(Y_x = 1 \mid x', z_1,z_2)$, and $P(Y_x = 1 \mid x', z_1,z_2,z_3)$. These are all exactly identifiable under *ctf-rand()*. This experiment concerns statistical error bars, not identification bounds.
> > >
> > > At fixed sample size ($N=200$), we showed that **the estimator variance increases as more variables are added to the query (and graph)**. This is expected: conditioning on more variables means estimating over finer sub-populations, which requires more data to maintain the same error bars. This experiment was included precisely to address the reviewer’s request for a practical discussion of what happens when more variables are introduced.
> > >
> > > ---
> > >
> > > > Could the authors discuss how practical is the method for real-world setups? For example, how does the method scale for a larger graph suppose 10-20 variables? What challenges will it face?
> > >
> > > As noted in the rebuttal, discovering the identification formula takes approximately $n^2(n+m)$, which is not a bottleneck even for graphs with $n>20$. **The main practical challenge is estimation once the formula is known.** Depending on the estimator, this step can be sample-inefficient or exhibit unstable convergence.
> > >
> > > To illustrate this concretely, we use the classic **Sachs et al.** dataset ([1], >5000 samples, 11-variable graph) and estimate an identifiable target query. We compare 3 estimation approaches using sub-samples of size 300:
> > >
> > > - **Naive MLE**: estimates each term in the ID formula separately and chains them together; very high variance.
> > > - **Bayesian sampling** [2]: lower variance, but computationally expensive and with less stable convergence.
> > > - **Neural causal models (NCMs)** [3]: somewhat more involved to set up, but low-variance and with stable optimization.
> > >
> > > In general, we have observed that NCMs offer the most favorable tradeoff among the 3.
> > >
> > > | Method | Error Range* | Wall-Clock Time |
> > > |---|---:|---:|
> > > | Naive MLE | 0.51 | 0:03 hrs |
> > > | Bayesian sampling | 0.18 | 5:27 hrs |
> > > | NCM | 0.11 | 0:17 hrs |
> > >
> > > *Error range calculated by re-estimating from different sub-samples of size 300.
> > >
> > > We believe that comparing unrelated real-world case studies at $n=10,15,20$ etc. would be difficult to interpret, because many factors change at once: the graph, the target query, variable arities, and the underlying data-generating process. Similarly, artificially hiding a fraction of nodes in one graph can abruptly introduce confounding or non-identifiability, making such scaling plots potentially misleading.
> > >
> > > ---
> > >
> > > **Note:** these findings concern *estimation*, which is **not** the main contribution of our paper. The significance of our central results are: (a) the completeness result (Thm. 2.1) shows that **any** sound method for this problem must be equivalent to the output of Algorithm 2; and (b) the impossibility results (Thm. 3.1, Cor. 3.2) show that **any** causal inference method must face these fundamental limitations. In this sense, *our theory is agnostic to the particular estimation pipeline used after the identification formula is derived.*
> > >
> > > ---
> > >
> > > **References:**
> > >
> > > [1] Sachs et al. (2005). *Causal Protein-Signaling Networks Derived from Multiparameter Single-Cell Data*
> > >
> > > [2] Zhang et al. (2022). *Partial Counterfactual Identification from Observational and Experimental Data*
> > >
> > > [3] Xia et al. (2023). *Neural Causal Models for Counterfactual Identification and Estimation*

---

### Decision · Program_Chairs · 2026-04-30

**Decision:**

Accept (regular)

**Comment:**

Reviewers agreed that the paper is well-presented, theoretically sound, and presents meaningful novel contributions in causal identifiability from counterfactual data that, while somewhat limited in scope/practical applicability, will be of interest to (some) causality researchers. Those concerns that were raised by reviewers were addressed and resolved by the authors' rebuttal, and the authors are urged to include the promised modifications and additions in the revised version. In summary, the paper will be a good addition to the conference and I recommend acceptance without hesitation.